# Development of an automatic delineation of cliff top and toe on very irregular planform coastlines (CliffMetrics v1.0)

Andres Payo[1], Bismarck Jigena Antelo[2], Martin Hurst[3], Monica Palaseanu-Lovejoy[4], Chris Williams[1], Gareth Jenkins[1], Kathryn Lee[1], David Favis-Mortlock[5], Andrew Barkwith[1], Michael A. Ellis[1]

[1]British Geological Survey, Keyworth, NG12 5GG, UK
[2]Cadiz University, Puerto Real, 11510, Spain
[3]University of Glasgow, East Quad, Glasgow, G12 8QQ, UK
[4]U.S. Geological Survey, Geology, Minerals, Energy and Geophysics Science Center, Reston, VA 20191, U.S.A.
[5]Environmental Change Institute, Oxford University Centre for the Environment, Oxford, OX1 3QY, UK

*Correspondence to*: Andres Payo (agarcia@bgs.ac.uk)

**Abstract.** We describe a new algorithm that automatically delineates the cliff top and toe of a cliffed coastline from a Digital Elevation Model (DEM). The algorithm builds upon existing methods but is specifically designed to resolve very irregular planform coastlines with many bays and capes, such as parts of the coastline of Great Britain. The algorithm automatically and sequentially delineates and smooth shoreline vectors, generates orthogonal transects and elevation profiles with a minimum spacing equal to the DEM resolution, and extracts the position and elevation of the cliff top and toe. Outputs include the non-smoothed-raster and smoothed-vector coastline, normals to the coastline- (as vector shapefiles), xyz profiles (as comma-separated-value files), and the cliff top and toe (as point shape files). The algorithm also automatically assesses the quality of the profile and omits low-quality profiles (i.e. extraction of cliff top and toe is not possible). The performance of the proposed algorithm is compared with an existing method, which was not specifically designed for very irregular coastlines, and to hand-digitized boundaries by numerous professionals. Also we assess the reproducibility of the results using different DEM resolutions (5 m, 10 m and 50 m), different user defined parameter-sets related to the degree of coastline smoothing, and the threshold used to identify the cliff top and toe. The model output sensitivity is found to be smaller than hand-digitized uncertainty. Code and a manual are publicly available on a github repository.

**Development and technical papers**

**1 Introduction**

Coastal cliff erosion is a worldwide hazard with impacts on coastal management, infrastructure, safety, coastal resilience and the local and national economies. Various types of cliffed and rocky coasts are estimated to represent about 80% of the world's oceanic shorelines (Doody and Rooney, 2015;Emery and Kuhn, 1982): these include plunging sea cliffs, bluffs backing beaches and cliffs fronted by rocky shore platforms. The increasing population of coastal zones has led to the accelerating

occupation of cliff tops and faces by buildings and infrastructure, including areas that are seriously threatened by shoreline retreat (Del Río and Gracia, 2009). The impact of this increased human presence has exacerbated erosion problems in some places. As a consequence, conflicts between human occupation and the inherent instability of cliffed coasts have become a problem of increasing magnitude (Moore and Griggs, 2002). Quantification of cliff retreat rates is vital for stakeholders who manage coastal protection and land use. An essential component of this quantification is reliable delineation of cliff location.

Automating the extraction of cliff top and cliff toe positions from topographic data will provide valuable constraints on coastal dynamics that will aid planning decisions, particularly where multi-temporal data are available, and thus will facilitate better predictions of coastal change. Cliff metric delineation has traditionally been done by hand-digitizing. Although efforts were made to standardize and eliminate subjectivity during hand-digitizing (i.e. Hapke et al., 2009), the delineation of cliffs and other shoreline features remains time-consuming and somewhat dependant on the analyst's interpretation.


**1.1 The problem of defining the top and bottom of a cliff**

Defining a cliff is a difficult problem. The top and bottom of a cliff are often readily apparent and implicitly defined along stretches of the coast with iconic vertical cliffs, e.g. those composed of chalk or massively bedded and indurate sedimentary rocks (Figure 1a). In less favourable circumstances, however, the relatively slow and sporadic erosion of cliffs may leave a

compound surface that (in profile and in plan view) is composed of partly concave and convex shapes. These compound surfaces may be further complicated by the occurrence of pre-existing and uplifted marine terraces, intervening coastal rivers (some of which may be hanging), and anthropogenic structures such as transport corridors (usually roads). These complications make the top of such a compound cliff profile difficult to define.

The situation is made more complex still when it is recognised that cliff erosion is, on a human time-scale, episodic. Cliff erosion occurs typically by land-sliding. Landslides are well known to be of a variety of types and, in general, their frequency and magnitude follow a power-law distribution (Hurst et al., 2013). This means that larger landslides occur much less frequently than smaller ones. An obvious and clearly visible "top" of the cliff may simply be the top of more frequent but relatively small landslides. Earlier and larger landslides may be visible in a topographic analysis of the sort described here, but

they may also be subsumed into anthropogenic landforms that form the boundaries of transport infrastructure (Figure 1b).

**Development and technical papers**

At the present day, cliffs are eroding in response to a relatively stable late Holocene sea-level established between 7 and 6 ky BCE, but the extent to which cliff erosion is accelerating or has reached a dynamic steady-state is also a function of the tectonic setting (e.g., whether land is actively uplifting or subsiding) and the form of the near-shore, which modulates wave energy as it propagates to the cliff line. Thus, the cliff top, if considered as the upper moving boundary of a dynamic process of cliff failure, is by no means easy to define: it may not always be the topographic high along the coast-normal profile. Still more complexity arises from the observation that cliffed coastlines are often interrupted by other non-cliffed coastal landforms such as estuaries and beaches (Figure 1c, d). For example, based on the European Commission (1998 – the CORINE project érosion cotière), the 14321 km of coastline of the British coast can be classified morpho-sedimentologically as: cliffs (67%) (56% hard-rock and 11% soft-rock), sand beaches (11%), shingle beaches (7%), heterogeneous beaches (4%) and muddy and estuarine coasts (10%) (May and Hansom, 2003).

The main advantage of an automatic algorithm for cliff top/toe delineation is that the uncertainty associated with hand digitization, which is subject to human error and subjective judgement, can be quantified and reduced. However, given the complexity of the problem, we acknowledge that this delineation will inevitably involve some ambiguity, which will only be resolved by human screening of the outputs. Therefore, a major requirement of any automatic cliff toe/top delineation procedure is some means of readily screening the outputs.

**1.2 Review of automatic delineation procedures**

Since cliff edges are linear features which are detectable in DEMs, automated and well-known methods used to extract breaklines can potentially be adapted to extract cliff edges. The automated methods of breakline extraction can be grouped into four major categories (Palaseanu-Lovejoy et al., 2016): (1) deriving lines from intersecting planes (i.e. Briese, 2004;Brzank et al., 2008;Choung et al., 2013), (2) extracting lines through a neighbourhood analysis of DEM elevation values (i.e. Rutzinger et al., 2012;Hardin et al., 2012;Mitasova et al., 2011), (3) applying edge detection filters and segmentation methods developed for image processing (Sui, 2002;Richter et al., 2013;Lee et al., 2009) and (4) automatic elevation profile elevation extraction analysis (Liu et al., 2009;Palaseanu-Lovejoy et al., 2016). The method that we present here belongs to the last category and its rationale is described below.

Liu et al. (2009) developed a method based on elevation profile extraction across the cliffs and the observation that generally the variation of the slope along the elevation profile is greater at the top and the toe of the cliff than anywhere else along the profile. However, this may not be the case for complex cliffs with roads or terraces cut through the cliff gradient, cliffs with different erosional profiles or slope gradients, or cliffs formed at the base of hills. Palaseanu-Lovejoy et al. (2016) (hereinafter PL2016) proposed an alternative method based also on profile extraction from high resolution DEM but that does not involve

variation in slopes between the profile point (i.e. cliff top and toe are delineated as the maximum and minimum respectively

of the de-trended profile). The PL2016 automatic delineation method has proven useful in resolving a range of types, from almost-vertical cliffs with sharply defined top and toe inflection points to complex cliff profiles. The PL2016 method relies on the user being able to generate a reference generalized vector shoreline which is free from tight bends and as much as possible is parallel to the general direction of the cliffs. The generation of the reference shoreline is however not part of the automatic delineation method itself. Such a generalized vector shoreline is not always possible to achieve for very irregular coastlines

(i.e. sequences of small bays and capes) such as parts of the northern and western coastlines of Great Britain. Also the length of the profile is a key parameter in this approach, but as shown by Palaseanu-Lovejoy et al. (2016), the method is robust enough that the position of the top and toe of the cliff does not change with the length of the profile, as long as the cliff is the most prominent geomorphic feature present. Ensuring that the cliff is the most prominent feature can be achieved by shortening/lengthening the profile length along the different coastline segments as done by PL2016 during pre-processing. But

even if the pre-processing is done carefully, it is likely that – due to natural variability of geomorphic features -- the cliff is not the most prominent feature in some locations. Thus this need to fine-tune the profile length for different coastal segments during the pre-processing stage detracts from the benefits of having an automatic delineation procedure. It remains unclear how the results might differ by using a fixed coastline normal versus a fine tuned normal length for each coastal segment.

Here, we present an automatic cliff toe/top delineation algorithm based on profile elevation extraction from a DEM, using a fixed profile length, and an automatic generation of a generalized coastline that is suitable for very irregular coastline shapes. The proposed method is demonstrated at several study locations along the British coastline using a DEM with national coverage. We compare the outputs of the proposed method with the outputs produced by the PL2016 method. We also explore the reproducibility of the results using different DEMs resolutions and user defined parameter settings (explained in detail

below). Model outputs are compared with the uncertainty of hand-digitized cliff toe and top as part of a sensitivity analysis of our approach.

Our software and documentation is available under Open Government Licence (see software availability section).

## 2 Study site and methods

**2.1 Digital Elevation Model source and study sites**

Our automated procedure requires a bare-earth Digital Terrain Model (DEM). The only requirement of the proposed method regarding the DEM is that it should include the cliff toe and top (i.e. cover from the shoreline to sufficiently far inland to capture the cliff top). The algorithm is agnostic regarding the method used to collect the data (i.e. air-prone-Radar, terrestrial/UAV LiDAR …). We have used several DEM from UK as an example of very irregular plan-shape coastline but

the method is in principle transferable to any other DEM. Here, we have used different resolutions of the NEXTMap DEM for Britain. NEXTMap for Britain is a 5m resolution DEM derived by airborne RADAR technology by Intermap Technologies. The elevation data was captured during 2002-3 and provides elevation point data on a 5 meters grid, which has subsequently been interpolated using a bespoke algorithm to derive the underlying 'bare earth' terrain model i.e. removing surface features such as buildings and trees. NEXTMap Height Data has a vertical accuracy of around 1m +/- RMSE and a horizontal accuracy

of 2.5m +/- RMSE on slopes less than 20 degrees. NEXTMap uses the OSGB36 Horizontal Datum and all elevations are relative to the Ordnance Datum Newlyn Vertical Datum. Radar cannot penetrate water and therefore the DEM records the elevation of the water surface at the time of image acquisition. Higher resolution DEMs of 10m and 50m where obtained by averaging the elevation of the 5m DEM.

The aims of the sensitivity, model-to-model comparison and hand-digitized analysis are different and therefore the places

selected to conduct each analysis are different too. Our sensitivity analysis and model-to-model comparison investigates the way in which the variation in the output can be attributed to variations in the different input factors (Pianosi et al., 2016) or different automatic delineation procedures respectively. The hand-digitized analysis illustrates the importance of the data outputs screening and algorithm behaviour. For the sensitivity analysis and model-to-model comparison, we have focused on a coastal cliff-dominated region with irregular plan-shape to make our findings more transferable to other similar cliffed

coastlines elsewhere. For the hand-digitized analysis we have selected a challenging coastal region (i.e. very irregular shape, complex cliff profile sections intercalated with non-cliffed sections) to highlight the importance of screening the results and running the algorithm iteratively until the hand-digitized and automatically delineated cliff top and toe locations converges.

For our sensitivity analysis, we selected a 30km coastal stretch centred at the St. Bees Head Heritage Coast in NW England.

This study area, which is part of the coast of the county of Cumbria, contains an assortment of different coastal morphologies but it is mostly dominated by high cliffs (Figure 2a). The southern section of the study area, south of St Bees Head, is fully exposed to the sea conditions from the Irish Sea, while the northern section is dissected by more sheltered estuarine environments. The rock has been eroded by wave action to produce the spectacular 80-metre high vertical cliffs stretching from the Seacote foreshore to Saltom Bay. At Fleswick Bay, a shingle beach laid on large sandstone platforms. At the west

end of the St. Bees valley, terminal moraines dating from the last glacial period (~12 - 14,000 BP) are exposed at the coast as bluffs. The West Pier at Whitehaven harbour forms a significant barrier to the movement of beach material further north. A small beach exists to the south of West Pier, formed by trapped beach material. The coastline for about 100 metres immediately to the north of Whitehaven harbour is protected by an armoured stone bank. A railway embankment fronts the natural cliffs along the coastline between Whitehaven and Harrington. At the northern limit of the study region is the port city of

Workington. Around Workington slag banks from blast furnace plants cover large sections of the coast, which also contains alluvial deposits from the river Derwent.

**Development and technical papers**

For the hand-digitized analysis we selected three 1 km sections that represent active cliffed coastlines of different height and plan shape – (Flamborough Head, North and South sections –  and one section that represents a non-active (i.e. Holocene) cliff

– (Sandhead (Figure 2b, c). The first cliffed section (FH1) was located on the south side of Flamborough Head, Yorkshire (UK), within highly erodable glacial tills deposited during Devensian glaciations (c. 35 to 11.5 ka BP).The second cliffed section (FM2) was located on the north face of Flamborough Head on the chalk cliffs, which are overlain by the glacial till deposits. On both sections FH1 and FH2, cliffs height are on the order of 20m but the coastline has a more irregular shape on FH2 than FH1. Section 3 (DG) is located near Sandhead, Dumfries and Galloway (UK) and is an inactive cliffed coastline. At

section DG, profiles maximum elevations are on the order of 20m.

**2.2  Automatic delineation of cliff metrics**

The automatic delineation procedure quantifies cliff top and cliff toe position, and cliff height, following the steps shown as a flow chart in Figure 3, illustrated further in Figure 4 and described in detail below. All the resulting geospatial outputs produced

by the proposed method are listed in Table 1.

    1.  Extracting the coastline from a DEM

        Figure 4a shows the input DEM that we use to illustrate the methodology. The first step is to delineate the shoreline at a user defined elevation. Coastline cells are delineated using a wall follower algorithm (Sedgewick, 2002). The wall is at the interface between cells above and below the user defined elevation. Raster cells "on" the shoreline are marked (Figure

4b); the coastline is also stored as a vector object. Depending on the coastal geomorphology and extent of each DEM tile, more than one coastline segment may be traced on the DEM. Each coastline segment is given an ID number ($N_{coast}$). The wall follower algorithm used to delineate the coastlines searches the tile edges to find the start of any coastline. The coastline of islands (i.e. land topography that does not cross the edges of the DEM) is not delineated (Figure 5). To resolve the islands the tile need to be zoomed-in to ensure that the edges of the land topography intersect any of the tile edges.


    2.  Generate a generalized (smoothed) coastline

        The resulting coastline is then smoothed to eliminate artefacts resulting from the resolution of the DEM, due to local geomorphic variability associated with the heterogeneity of natural landscapes, and the presence of man-made features at the coast, in order to produce a generalized coastline. This is done either by running a moving average window across the positional

X-Y coordinates, or by Savitzky and Golay (1964) smoothing, which involves fitting successive sub-sets of adjacent data points with a low order polynomial using least squares regression. The user need to decide which method better fit his perception of a generalized coastline. The resulting smoothed coastline comprises a compound vector object. This is made up of two set of consecutive points: a set holding the location of each smoothed coastline point and a set holding the original non-smoothed cell location of the coastline point (Figure 4c).

**Development and technical papers**

3. Extract transects normal to the coast

We then generate cross-shore transects, from which we extract the coastal topography. These cross-shore transects are located perpendicular to the smoothed coastline, extending inland from each coastline cell for a user-defined distance (Figure 4d). Normals that intersect the coastline more than once (e.g. barrier beaches, headlands) are flagged as "hitting the coast" profiles; their length is reduced (i.e. the profile is shortened to the segment between the first and the second shoreline intersection). Coastline normal transects that are too short (i.e. extend for only two raster cells) are considered invalid for the delineation of the cliff metrics: these are flagged as "non-valid". Intersecting coastline normals are flagged as "intersecting but not truncated".

4. Morphometric identification of the cliff top and toe

Coastline-normal elevation profiles are then sampled from the DEM cells under each valid coastal normal. The elevation of each point of the coastline normal is determined using the elevation of the centroid of the closest raster cell (thus coarser resolution DEMs will produce more jagged elevation profiles). A topographic trend line is then calculated as the elevation difference between the start and end points on the profile, divided by the horizontal profile length (Figure 4e). The de-trended profile elevation is then calculated as the residual when the topography is compared with the trend line. Finally, the cliff top and cliff toe are identified as the maximum and minimum de-trended elevations (Figure 4f). All cliff toe and top points for the input DEM, as identified using this procedure, are shown as a 3D model in Figure 4. Note that an optional user-defined elevation threshold may be used to avoid false peaks. If the absolute value of the peak elevation/depression is lower than the threshold elevation, it is assumed that the points at the end/start of the profile are the cliff top/toe respectively. This 'elevation sanity check' is required to avoid small bumps on rather slope-uniform profiles (i.e. non-cliffed coastlines) being picked up as cliff tops/toes.

**2.3 Sensitivity analysis and model vs model comparison**

We assessed output sensitivity to: (1) the DEM resolution, using DEMs of 5m, 10m and 50m of the same study region, (2) the degree of smoothing of the generalized coastline, and (3) the threshold used to avoid false cliff top/toe locations. Table 2 summarizes results from these sensitivity analyses. As a reference, we used the cliff metrics outputs for the DEM of 5m resolution, a 61-cell moving average window for coastline smoothing, and 0.5m as the vertical threshold. This distance seems to be large enough to produce a smooth coastline and small enough to resolve the numerous headlands and bays along this part of the British coastline. To explore the local sensitivity to coastline smoothing, we have also used 31 cell and 7 cell moving

average window size that are equivalent to ~165-220 m and ~35-50 m windows respectively. We selected the vertical threshold of 0.5m as the reference threshold because it is of the same order of magnitude of the vertical accuracy of the RADAR elevation data used by NEXTMap. The reference threshold elevation is relative to the de-trended elevation and it can be smaller than

the DEM resolution. To explore the local sensitivity to vertical threshold value, we also used vertical thresholds of 0.01m and 1.5m.

Figure 6 shows the smoothed coastline obtained, using these different DEM resolutions and different smoothing window sizes, for six different coastal morphological environments: (a) estuarine, (b) bay with harbour, (c) un-interrupted high cliffed coast line, (d) pocket beach surrounded by high cliffs, (e) beach at the seafront of a relic valley, (f) low cliff coastline (i.e. eroding

moraines).

When the number of points for the window size is chosen to make the window length similar under different DEM resolutions, the resulting smoothed coastlines are very similar. In particular, the smoothed coastlines for the 5m DEM and 61 pints and the 10m DEM and 31 points are almost identical. In all cases, the smoothed coastline differs from the high water line, which is

expected when using a still water level of 1.0m above the OD to delineate the coastline. By choosing a water level of 1.0m above Ordinance Datum, we have avoided delineating man-made coastal infrastructure, such as the Whitehaven harbour, which elevation has not been fully removed from the DEM. Around the Workington harbour, the estuary cuts the edges of the DEM and the model automatically creates two coastlines (a short one-north side of the Workington harbour, and a longer one to the south).


Choosing metrics to compare model outputs is not straightforward. The number of cliff top/toe points varies with the DEM resolution (because. the method delineates one coastline normal through every coastal cell point) making a profile-to-profile comparison infeasible (because profile elevation and orientation will also vary with DEM resolution and selected coastline smoothing). Thus, we chose a point-to-line-distance approach. Points are the cliff top/toe location outputs; as a reference line,

we converted the cliff toe/top points into a cliff top/toe line, for the reference model set up. The minimum distance between the cliff toe/top locations and the reference line was calculated using the QGIS 2.18.3 "Distance to nearest hub" tool. Given a layer with source points (i.e. cliff toe/top points) and another layer representing destination points or lines (i.e. reference cliff toe/top line), this "Distance to nearest hub" tool computes the distance between each source point and the closest destination one. The shortest distance between any point and a line is the length of the line segment that joins the point to the line and is

perpendicular to the line. We calculated the average, standard deviation, maximum and minimum shortest distances for all source points.

For the model-to-model comparison, we compared the model outputs for the reference set up with the PL2016 model outputs. Both methods differ regarding the pre-processing that is required (Table 3). The PL2016 method requires more pre-processing

than our approach since PL2016 needs to create a generalized coastline, split the coast into segments, and then associate a buffer width with each segment. For the St Bees study region of circa 30km, the coastline was divided into 25 segments. Buffer width ranged from 20 m to 400m (Figure 7). Our method delineates the smoothed coastline automatically and does not require the coastline to be divided into segments. However, coastline segments will be created if the delineated coastline cuts the edges of the DEM domain. We used the smoothed coastline produced by our algorithm, using the reference set-up, as the generalized

line required for the PL2016 method. Both methods are therefore quite similar with regard to coastline selection. The main difference concerns the way that the coastline normals are defined. After some trial and error, we chose a profile length of 500m as our user defined fixed length. As a metric of the differences in outputs, we again use the QGIS "Distance to nearest hub" to calculate the differences on the cliff top and toe locations outputs produced by the method proposed here and PL2016.

**2.4 Hand digitized profiles analysis and iterative output screening method**

As outlined in the introduction section, a major requirement of any automatic cliff toe/top delineation procedure is some means of readily screening the outputs. In this section, we describe how we have developed a methodology to iteratively screen over the model results and run the automatic delineation algorithm to achieve a desired model behaviour or identify any bias on the target lines.


The target cliff top and cliff toe locations are obtained from a cluster of 24 hand-digitized lines from aerial photography. A group of 24 participants with a range of geological expertise participated in the experiment, each interpreting data for three 1 km sections (FH1, FH2 and DG, see Figure 2b, c). Using a Geographic Information System (GIS) (Google Earth Pro 7.1.8.3036 (32-bit)), participants attempted to delineate cliff top and toe lines without any prior knowledge of their location. As with the

sensitivity analysis, we used a point-to-line metric to calculate the main statistics of the hand-digitized results. As a reference line, we generated a mean cliff top and toe line for each of the study sections from the participant data. We extracted the cliff top and toe points from each one of the hand-digitized lines and calculated the average, standard deviation, maximum and minimum shortest distances for all source points. This provides us with both, a quantitative assessment of the uncertainty in human interpretation of cliff top and toe lines from aerial photography as well as a number of target cliff and top and toe lines

to test the proposed algorithm behaviour.

Building on; (1) the uncertainty on the human interpretation of cliff top and toe lines from aerial photography, (2) sensitivity analysis results and (3) model outputs (see Table 1) we developed an iterative output screening method to achieve the desired model behaviour and identify bias on the target lines. We clustered the hand-digitized lines to broadly capture the different

interpretation of coastal cliff toe and top. The different clusters were then linked to the different model set up parameters. We then illustrate a model output screening method and iterative parameter selection for users to achieve desired model behaviour.

## 3 Results

**3.1 Output sensitivity to DEM resolution, coastline smoothing and vertical threshold**

Figure 8 shows the cliff toe and top locations for a high-cliffed coastal segment at the St. Bees study site for different DEM resolutions and for different smoothing window sizes. The cliff metrics for the 5m DEM-61 points window size and 10m DEM-31 points window size are very similar and are clearly different to the metrics obtained for the 50m DEM-7 points window size. The cliff metrics for the 3D models illustrate how the cliff top and toe locations relate to the resolution of the

DEMs. Figure 9 shows the cliff metrics for all six regions and the 3D model derived from the 5m DEM. While our approach is designed to resolve cliffed coastlines, it also seems to be able to resolve very irregular coastline shapes such as the one of a pocket beach between high cliffs (d), bay (b), and estuarine (a) environments.

Table 4 shows the results for the cliff-toe sensitivity analysis for the St. Bees study case. The average difference between the

cliff-toe location outputs and the reference outcome varies between 1m to 26m. It is most sensitive to changes in DEM resolution (i.e. average differences of 4m and 25m for the 10m and 50m DEM resolution, respectively). Cliff-toe location is less sensitive to changes in the size of the smoothing window and the vertical threshold (i.e. differences always smaller than 2m). Standard deviation is largest (about 40m) for the DEM of 50m resolution, and isabout 10m for the outputs from the DEM of 5m and 10m resolution. The maximum difference is 368m for the DEM of 50m and vertical threshold of 1.5m.


Table 5 shows the results for the cliff-top sensitivity analysis. Average differences between the cliff-top location outputs and the reference outcome vary between 0m to 37m, again being most sensitive to changes in DEM resolution (i.e. average differences of 6m and 32m for the 10m and 50m DEM resolution, respectively). Cliff-top location is (again) less sensitive to changes in the size of the smoothing window and the vertical threshold (i.e. differences always smaller than 8m). Standard

deviation is largest (about 60m) for the DEM of 50m resolution, and is about 10m-20m for the outputs from the DEM of 5m and 10m resolution. The maximum difference is 502m for the DEM of 50m and 7 points smoothing window size. There are about 6500 coastal points, 3200 points and 600 points for the DEMs of 5m, 10m and 50m resolution.

Since model outputs are most sensitive to DEM resolution, we extended the sensitivity analysis to DEM resolutions of 15m,

20m, and 35m. To keep the window-size-length of similar magnitude, we chosen the window size for smoothing the coastline to be 21, 15 and 9 points for the 15, 20 and 35m resolution DEMs respectively. We kept the vertical threshold unchanged (0.5m). Figure 10 shows average differences decreasing as the DEM resolution decreases. To estimate the trend in average

differences, we fitted and extrapolated a polynomial line of order 3 to the cliff top and toe calculated differences. This fitted trend line suggests that the minimum differences (i.e. for the smallest DEM resolution) are 1m and 5m for the cliff toe and top

respectively.

## 3.2 Model to model comparison

Our results show that the two automatically-delineated cliff top and toe locations are, generally, in good agreement (i.e. distances are less than one cell-diagonal). Toe locations are anticipated to be different since the proposed method uses a user-

defined elevation (e.g. 1.0 m: chosen to avoid delineating the man-made infrastructures near the coast which had not been removed from the DEM) to begin its coastline profiles, while the PL2016 method begins its transects from the lowest elevation (i.e. 0 m for the DEM used here). Distances between cliff metrics of less than one cell-diagonal length (i.e. 7.07m for a 5m cell size) are considered within the DEM resolution limit and thus, for model-to-model comparison purposes, identical outputs. The PL2016 method applied to the St. Bee study site produced a set of 6655 toe points and 6324 top points (i.e. top points are

less than toe points because concave profiles are not used to delineate the cliff top but the profile is still been used to delineate cliff toe). Our approach produced a data set of 6598 top and toe points, of which 68 were flagged as poor-quality points. The minimum distance between the line formed by the proposed method cliff top and toe outputs and the PL2016 outputs was calculated: the frequency distribution of the minimum distance between the cliff top and toe locations resulting is shown in Figure 11. The cliff toe locations are in good agreement (i.e. minimum distance less than one cell-diagonal) for 78% of the

data, and the cliff top locations are in good agreement for 68% of the locations. The median distance for both top and toe locations always inferior to a cell-diagonal. The mean distance value is larger than the median, and a cell-diagonal, and of 8.41m and 10.93m for the toe and top locations. To understand where the outputs from the two methods differ between each other, it is necessary to look at the spatial distribution of the differences.

Figure 12 shows the spatial differences between the cliff metrics results using the PL2016 algorithm and our approach. The maximum differences in toe (236m) and top location (206m) are either in segments where the coastline makes sharp bends and the cliff is not the dominant feature (Figure 12a), or where the cliff has a steep face with talus at the toe (Figure 12b). Both methods were able to delineate the cliff metrics along the eroding moraines, but our approach was also able to trace a welded bar at the southern end of St. Bees beach (Figure 12c). The bar crest elevation is of the order of 2m height. It lays parallel to

the coastline with eroding moraines of approximately 15m height. Most of the sea-facing cliff toe and toes along the bar were flagged as non-valid. At the inland-facing side of the bar, most of cliff toe and tops were flagged as valid (i.e. long enough and top elevation higher than toe).

**3.3 Hand digitized uncertainty**

Analysis of the resulting violin plots for each location (Figure 13) reveal that there is less variance in defining cliff toes when compared to the cliff tops, and the results are skewed towards the seaward side of the mean delineations. The lowest range in cliff top delineation comes from the FH1 site, where there is a 32.16 m spread between the 25th and 75th percentiles. The largest spread in cliff tops comes from the DG site, where there is a 166.97 m spread in the same percentiles. Further analysis

of each section shows that two distinct peaks, separated by over 100m, are present in the derived histograms for each cliff top site. The spread of delineations around these peaks is similar to those for the cliff toes. The smaller range in cliff toe-line variance suggests that there is greater certainty in participants defining those lines from aerial photography. The negative skew within the violin plot analysis is likely due to tidelines, beach and platform being readily identifiable in the images and therefore less prone to be misinterpreted as cliff-line features. The bi-modal nature of the cliff top delineation can be attributed to

participants' personal definition of what constitutes a cliff top. This dilemma is highlighted in the FH1 and DG sites. In the former there are two distinct breaks in slope and participants tended to follow either a higher or lower cliff top line (Figure 14). This dilemma is not present on the FH2 site were only one distinct break in slope exists (Figure 15). Within the DG site there is a very low cliff (<1 m) at the top of the beach and a much more pronounced Holocene cliff line set back around 100m from the coastline (Figure 16). Participants tended to prefer either one cliff line or the other. Interestingly for the DG site, even

if participants selected the Holocene cliff-top, they were unlikely to use the Holocene toe-line. This is highlighted by the lack of bimodal response in the DG toe line histogram.

**Error! Reference source not found.**

Figure 17 shows the automatically delineated cliff top and toe for the DG and FH sites using different input model set up. Starting with the same model setup used as a reference for the sensitivity analysis (DEM of 5m resolution, a 61-cell moving

average window for coastline smoothing, and 0.5m as the vertical threshold), and simply changing the still water level used to delineate the coast line from 0.01m to 6m and changing the profile length from 105m to 500m the algorithm is able to differentiate between the active cliff profile (still water level = 1m & profile length = 105m, Figure 17a) and the Holocene cliff (still water level = 6m & profile length = 500m, Figure 17b). By rising the still water level, we obtain generalized coastlines that represent current mean sea level and raised historical sea levels. By using a smaller profile length for the active profile we

ensure that, the active cliff is the dominant feature captured. At the location where the Holocene cliff is very close to the active cliff, the algorithm pick up the highest Holocene cliff as the dominant cliff feature but at the right and left sides picked up the active cliff. The reference model input (DEM of 5m resolution, a 61-cell moving average window for coastline smoothing, and 0.5m as the vertical threshold, still water level 1m, profile length 500m) seems to provide reasonable locations of cliff top and toe at FH1 and FH2 sites (Figure 17c, d). From the FH sites it seems clear that the automatically delineated cliff top does

not everywhere corresponds with an abrupt change of slope.

## 4 Discussion and conclusion

Cliff metric delineation has traditionally been done by hand-digitizing. Although efforts were made to standardize and eliminate subjectivity during hand-digitizing (i.e. Hapke et al., 2009), the delineation of cliffs and other shoreline features remains time-consuming and somewhat dependant on the analyst's interpretation. PL2016 proposed method based on profile

extraction from high resolution DEM that has proven useful in resolving a range of types, from almost-vertical cliffs with sharply defined top and toe inflection points to complex cliff profiles. However, the PL2016 method relies on the user being able to generate a reference generalized vector shoreline which is free from tight bends and as much as possible is parallel to the general direction of the cliffs. The generation of the reference shoreline is not part of the PL2016 automatic delineation method itself. Such a generalized vector shoreline is not always possible to achieve for very irregular coastlines (i.e. sequences

of small bays and capes) such as parts of the northern and western coastlines of Great Britain. Also the length of the profile is a key parameter in this approach, but as shown by Palaseanu-Lovejoy et al. (2016), the method is robust enough that the position of the top and toe of the cliff does not change with the length of the profile, as long as the cliff is the most prominent geomorphic feature present. Ensuring that the cliff is the most prominent feature can be achieved by shortening/lengthening the profile length along the different coastline segments as done by PL2016 during pre-processing. But even if the pre-

processing is done carefully, it is likely that – due to natural variability of geomorphic features -- the cliff is not the most prominent feature in some locations. Thus this need to fine-tune the profile length for different coastal segments during the pre-processing stage detracts from the benefits of having an automatic delineation procedure. Until now, it was unclear how the results might differ by using a fixed coastline normal (Figure 6c) versus a fine tuned normal length for each coastal segment. Here, we have presented an automatic cliff toe/top delineation algorithm based on profile elevation extraction from a DEM,

using a fixed profile length, and an automatic generation of a generalized coastline that is suitable for very irregular coastline shapes. The proposed method is demonstrated at several study locations along the British coastline using an air-prone-radar DEM with national coverage at different resolutions. The algorithm is agnostic regarding the method used to collect the DEM and therefore it could be applied to other methods such as UAV/drone/terrestrial elevation data collection procedures. The main differences and similarities between the two methods are summarized in Table 3.

Fine-tuning the profile length, as proposed by PL2016, makes an appreciable but small difference to cliff toe and top automatic delineation when using a fixed profile length. The comparison of the outputs produced by the proposed method, which uses a fixed profile length, and the PL2016 method, which fine-tunes the profile length for each segment along the coastline, suggest that cliff toe and top location are virtually the same for more than 2/3 of the cases. For those cases where outputs location do differ, neither method seems to outperform the other. By avoiding the need of fine-tuning the profile length, the proposed

method speed up the delineation process but does not eliminate the need of the screening of the model outputs.

To facilitate the screening of the model outputs, our approach produces a set of shape and ASCII files (Table 1). These output files are therefore in a format that is readable by most GIS and spreadsheet software (i.e. QGIS, Excel,…). These outputs are

an important requirement of any automatic delineation procedure. They are labelled in a self-explanatory fashion to allow the
user explore the underlying data of any delineated cliff top/toe location.

Hand-digitized cliff top and toe location spread between participants are of the order of 4 to 23 diagonal cells (i.e. for a DEM of 5m cell size). These large differences seems to be driven by the bias towards using changes of slope as the preferred cliff top and toe locations when hand digitizing over an aerial photography. This bias prevented a model to hand-digitized more in
detail comparison. We have shown how the inputs parameter can be modified to resolve both active and Holocene cliff lines. The algorithm reference set up seems to be robust enough for the two FH sites despite the difference on the plan shape at both sites. Our algorithm delineates the cliff top and toe and produce all model outputs for a 1km section of coast in less than one second while hand-digitizing the same length of coast took around 10 minutes. Thus our algorithm is about 5 orders of magnitude faster than hand digitizing.


To conclude, we developed and demonstrated a new automatic delineation procedure of cliff toe and top location based on the extraction of profile elevation from a DEM. This approach requires less pre-processing than other existing automatic methods, and it facilitates the screening of the delineated locations by outputting key supporting information. Our approach will be of great value in tracking changes of cliff metrics along coastlines of irregular shape.

**Code availability**

The code for the proposed automatic delineation of the generalized coastline and cliff metrics has been coded in C++ and the source code is available at https://doi.org/10.5281/zenodo.1412486. See https://github.com/coastalme/CliffMetrics for the latest version of the source code. CliffMetrics builds easily using Linux. The CliffMetrics code uses the open-source GDAL package (version 2.1.3) to read and write the shapefiles and raster files. For this study, CliffMetrics was compiled using gcc
435     4.8.4.

To install and run CliffMetrics under Linux:
1. Create a local copy of the github repository, for example by downloading a zipfile, then unpacking it
2. At a command-line prompt, change to the CliffMetrics master folder, then to the src folder
3. Run_cmake.sh. If you see error messages re. missing software (for example, telling you that CMake cannot be found or is too old, or GDAL cannot be found or is too old) then you need to install or update the software that is causing the problem
4. Run make install. This will create an executable file called cliff in the CliffMetrics-master folder.

**Development and technical papers**

5.  Edit cliffmetrics.ini to tell CliffMetrics which input file you wish to use (for example, in/Example/UserInputs.dat). The user inputs data file contains the user defined delineation parameters (Table 6).

     6.  Run cliff. Output will appear in the out/ folder.

**Acknowledgements**

This work was funded by the Natural Environment Research Council (NERC) as part of the (BLUEcoast) project
(NE/N015649/1).

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

**Development and technical papers**

**Figures & Captions**

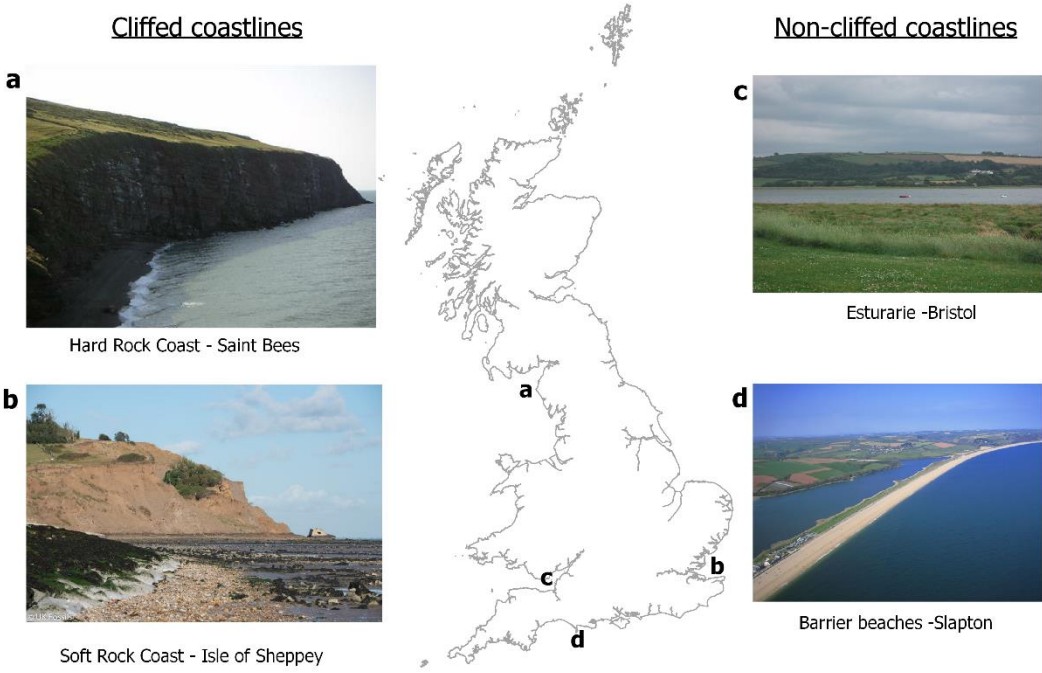


Figure 1: The problem of defining the top and bottom of a cliff is not trivial. For example, most of Britain coastline is made of cliffs (hard and soft) but also beaches and estuarine environments. (a) Cliff top and toe are readily apparent for the hard rock coast of Saint Bees but not as clear at the soft cliffed coastline of the Isle of Sheppey where landslides are ubiquitous (b). Cliffed coastlines are often interrupted by other landforms such as estuaries (c) and beaches (d).

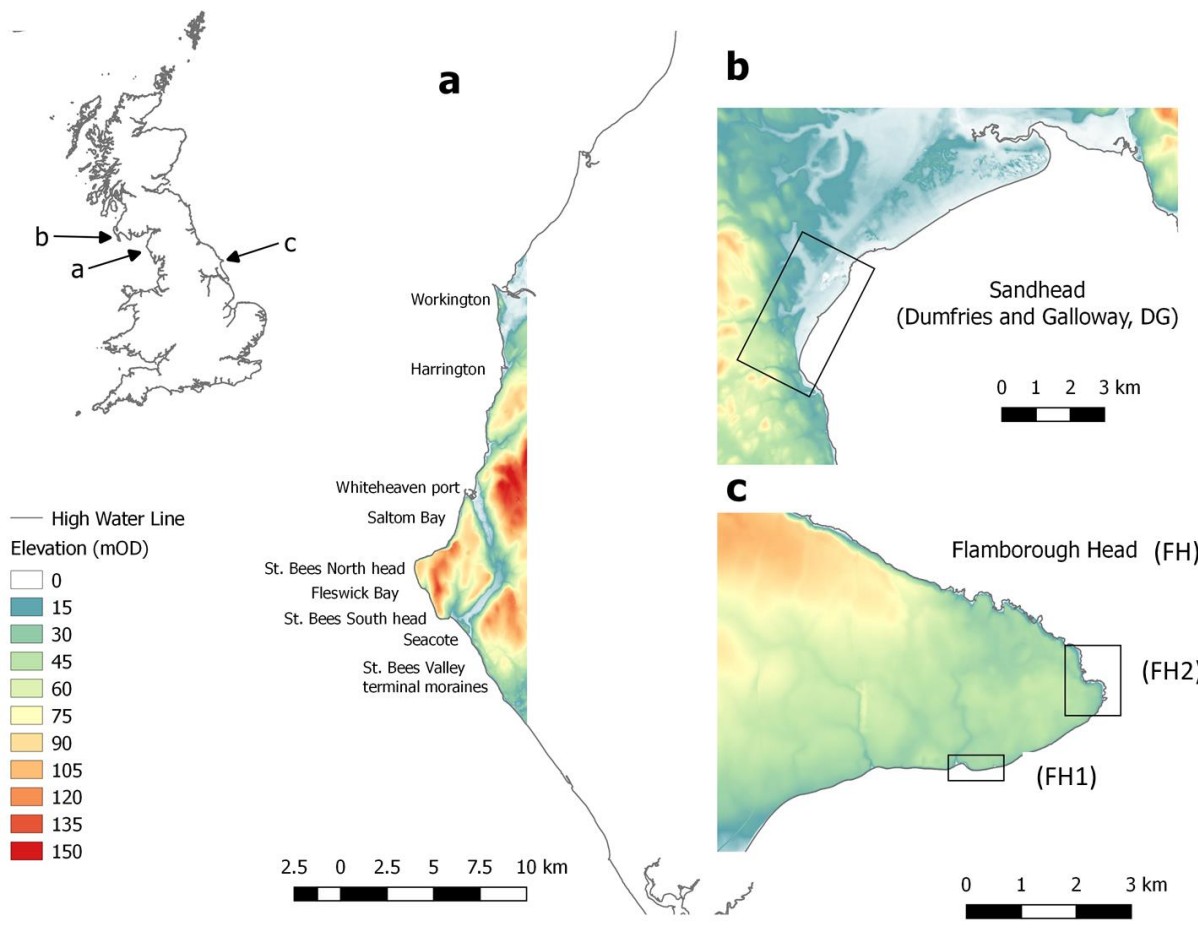


**Figure 2: The NEXTMap DEM of selected study sites around Britain's coastline; (a) St. Bees head in NW England, used for the model sensitivity analysis. The name of the main locations cited in the text are shown along this coastal stretch, (b) Sand Head and (c) Flamborough head sites are non-active and active cliffed coastlines sites used for the hand-digitized uncertainty analysis. At Flamborough, two study sites were selected with cliffs of similar heights but with relative uniform coastline (FH1) and very irregular coastline shapes (FH2).**


**Development and technical papers**

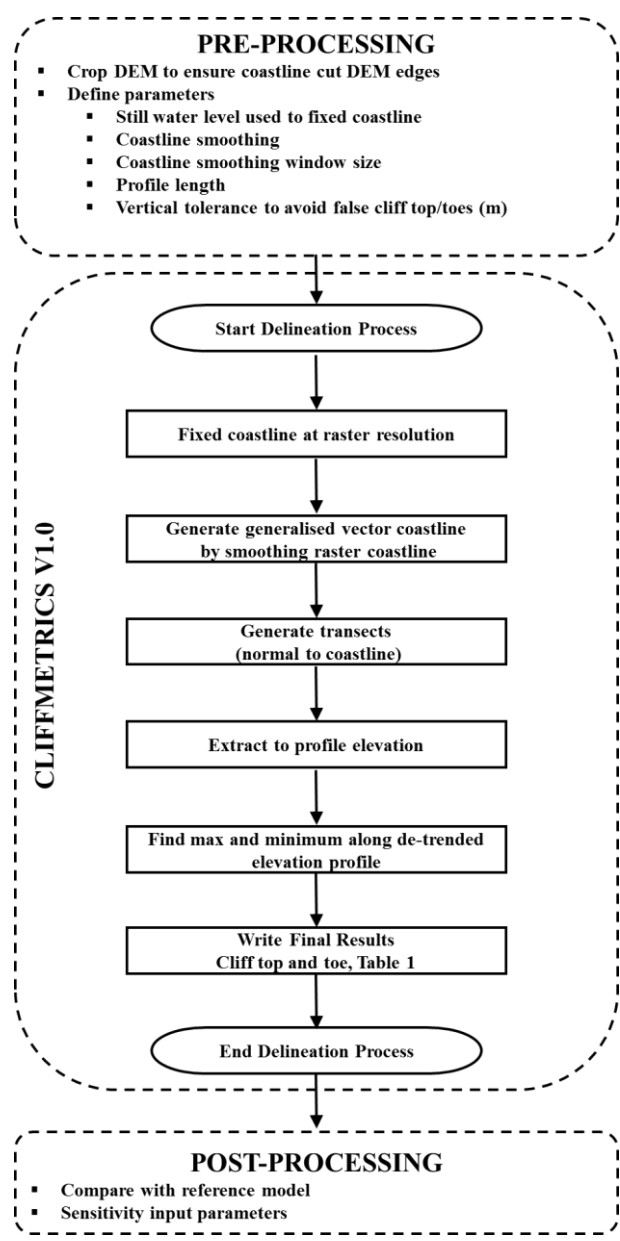

**Figure 3. Flow chart of the proposed automatic delineation algorithm**


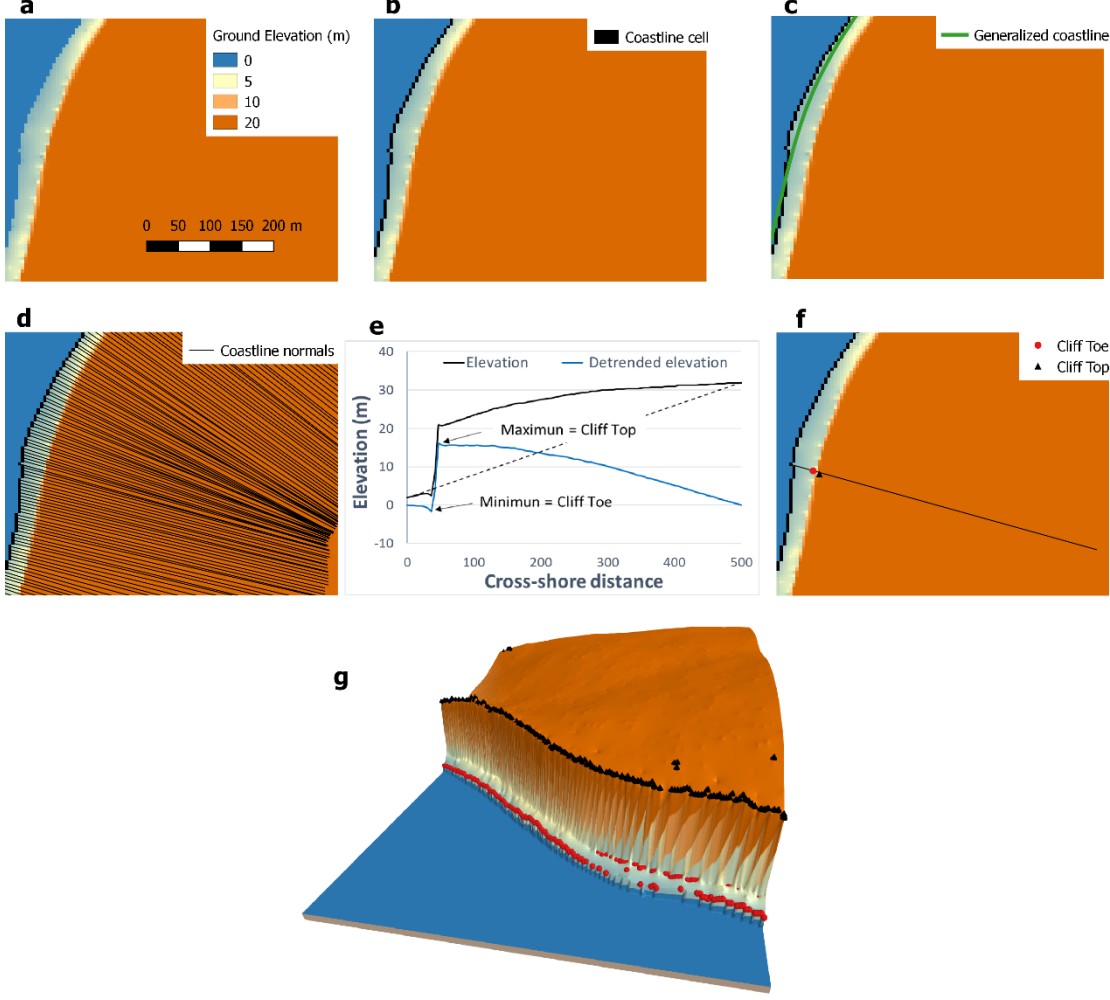

**Figure 4: Illustration step by step of the proposed method to generate the generalized coastline and extract the cliff toe and top elevation and location; (a) the input DTM (b) the cells on the coastline are marked and (b) smoothed to create a generalized coastline vector, (d) coastline-normals are delineated starting at the cells marked as on the coastline and perpendicular to the straight line connecting the before and after smoothed-coastline point, (e) profile elevation is extracted along each normal and cliff top and toe are located as the maximum and minimum elevation of the detrended elevation profile; (f) shows the location of the cliff top and toe along the elevation profile shown in panel (e); (g) shows the DTM in 3D and the outputs location of cliff toe (red circles) and top (black circles).**


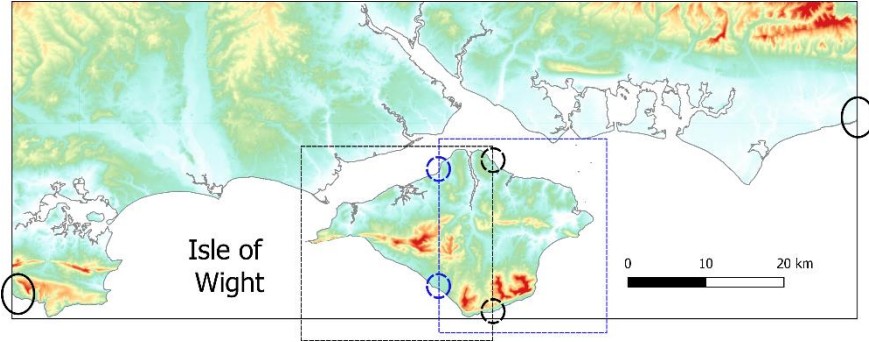

**Figure 5: The algorithm used to delineate the coastline searches the edges of the tiles to find the start point of the coastline. The solid grey line represent the coastline over the coloured DEM (i.e. the hotter the colour the highest the elevation). The coastline cut the edges of the DEM at the locations indicated by the solid black circles. The coastline of the Isle of Wight does not cut the edges of the DEM and therefore the user needs to define two smaller DEM domains (represented as dashed black and blue rectangles for the West and East side of the Isle). The Isle coastline does now cut the smaller domains at the locations indicated by the blue and black dashed circles. It is recommended to allow some overlap between the smaller domains to ensure that the cliff metrics is well resolved near the edges.**

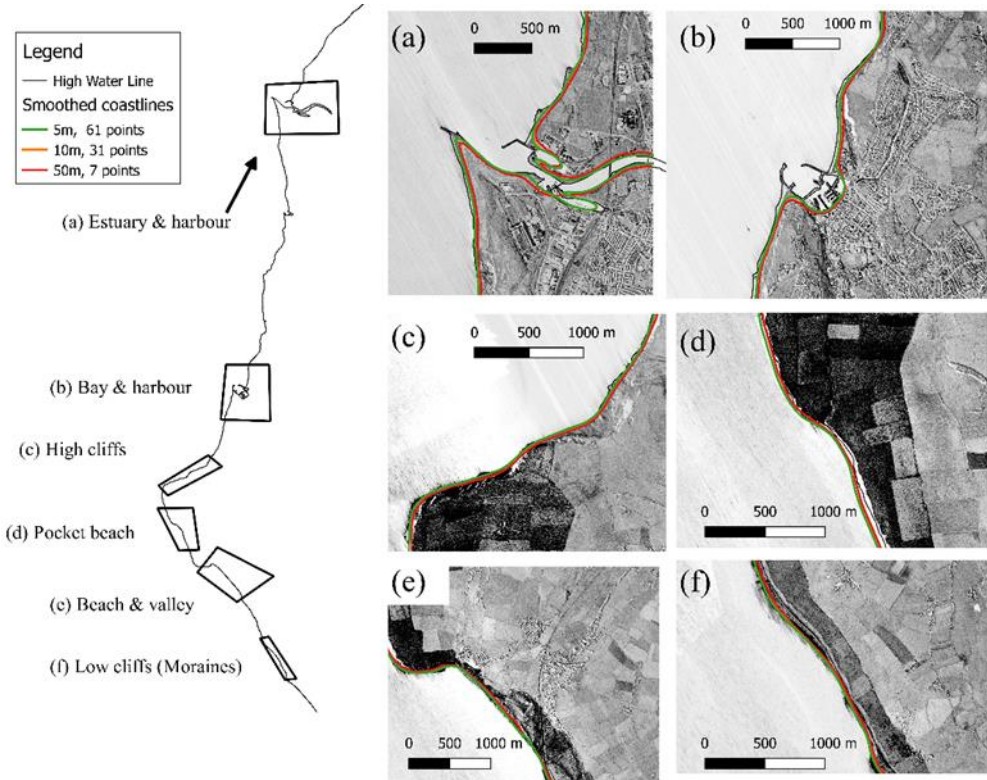

**Figure 6: Location of six different coastal morphologies around St. Bees Heritage coastline and smoothed coastline obtained for different DEM resolutions and window size used for smoothing. Original RADAR images used to build the NEXTMap DEM shown on grey scale. Smoothed line for the 10m DEM with 31 points window size (orange line) is almost identical to the line obtained for the 5m DEM with 61 points window size (green line) and not always visible.**

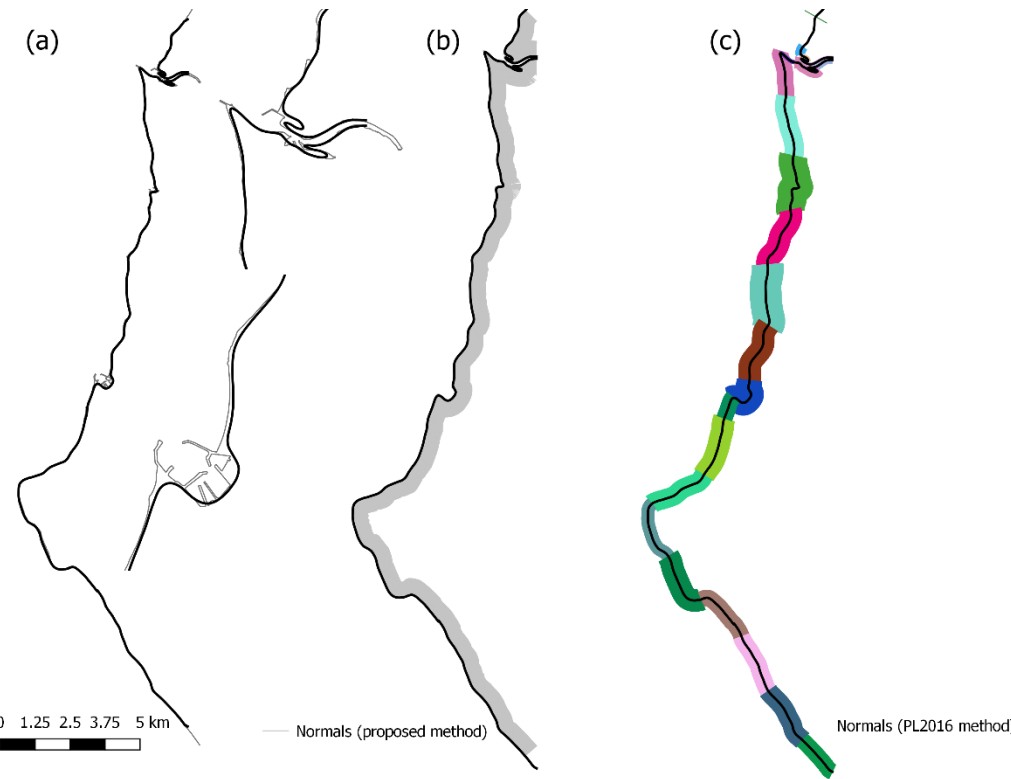

**Figure 7: Generalized coastline and coastline normals used for the proposed method and the PL2016 method. (a) Smoothed coastline (solid black) on top of high water line (solid grey). Zoom in around Whitehaven and Workington harbours illustrate the differences between both lines. (b) Coastline normals derived using the proposed methodology. (c) Coastline normals derived using PL2016. The different colours represent the different segments used.**


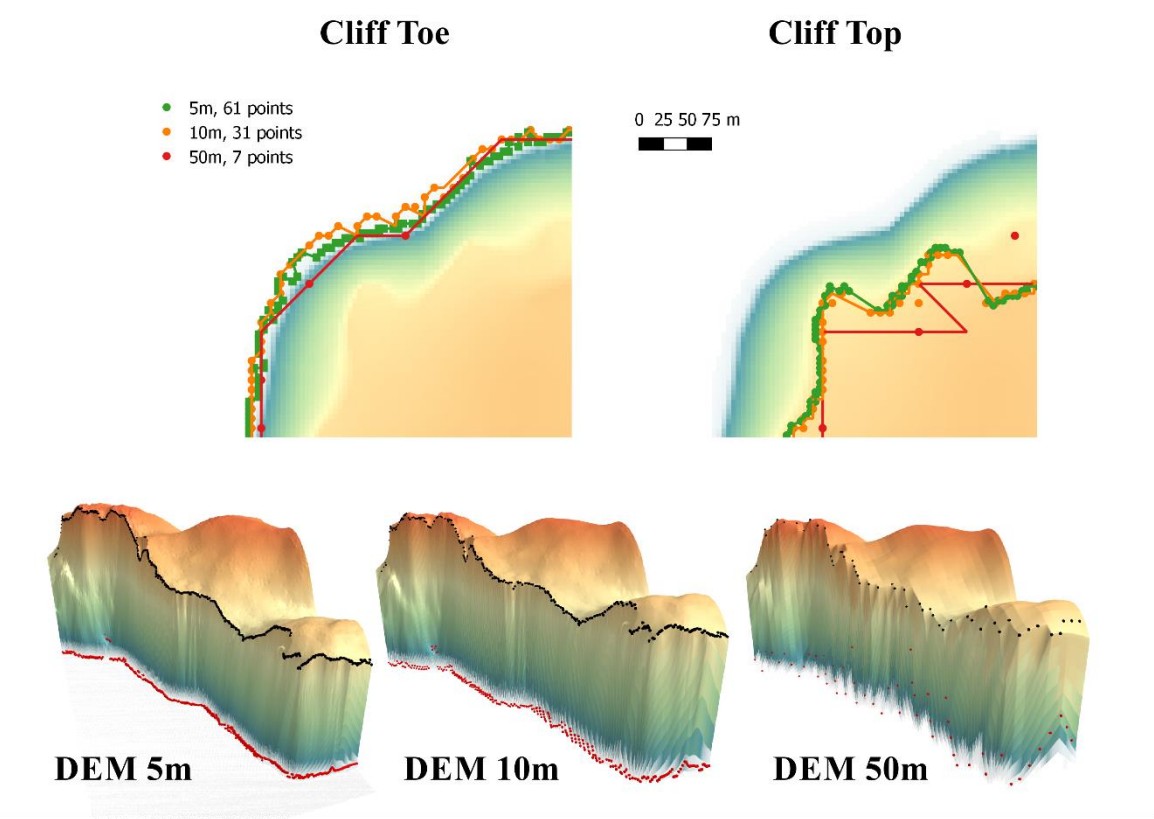

**Figure 8: Cliff toe and top outputs for a high cliffed coastline section and different DEM resolutions and window size smoothing window. Upper panel shows the location of the cliff toe (left) and cliff top (right). Bottom panel shows a 3D model with the cliff toe and top as red and black spheres. Vertical dimension of 3D model has been exaggerated 10 times.**


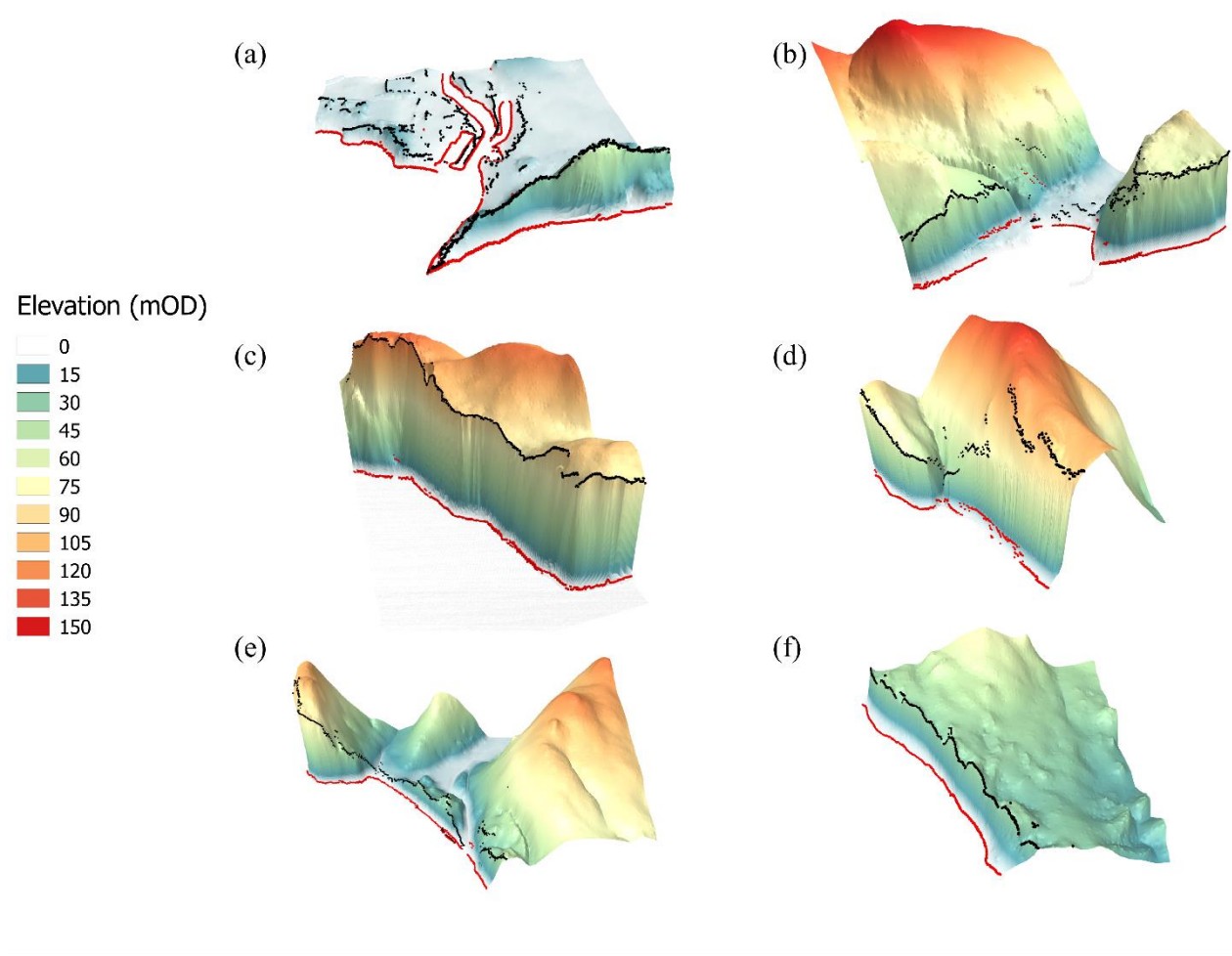

Elevation (mOD)

| | |
|---|---|
| | 0 |
| | 15 |
| | 30 |
| | 45 |
| | 60 |
| | 75 |
| | 90 |
| | 105 |
| | 120 |
| | 135 |
| | 150 |

**Figure 9: 3D models of different coastal morphology environments with the cliff toe (black spheres) and cliff top (red spheres) delineated using the 5m DEM and 61 points smoothing: (a) estuarine, (b) bay with harbour, (c) high cliffed coastline, (d) pocket beach, (e) beach at the seafront of a relic valley, (f) low cliffed coastline (eroding moraines).**

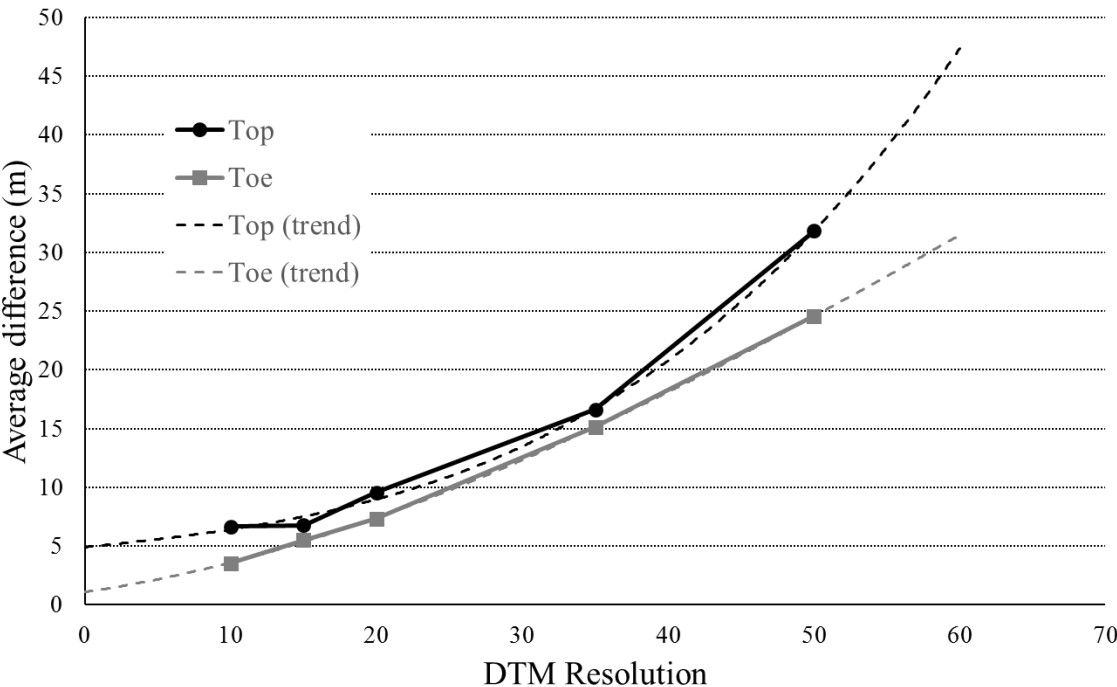


**Figure 10: The cliff metrics outputs (top and toe location) average difference decreases as the DEM resolution decreases.**

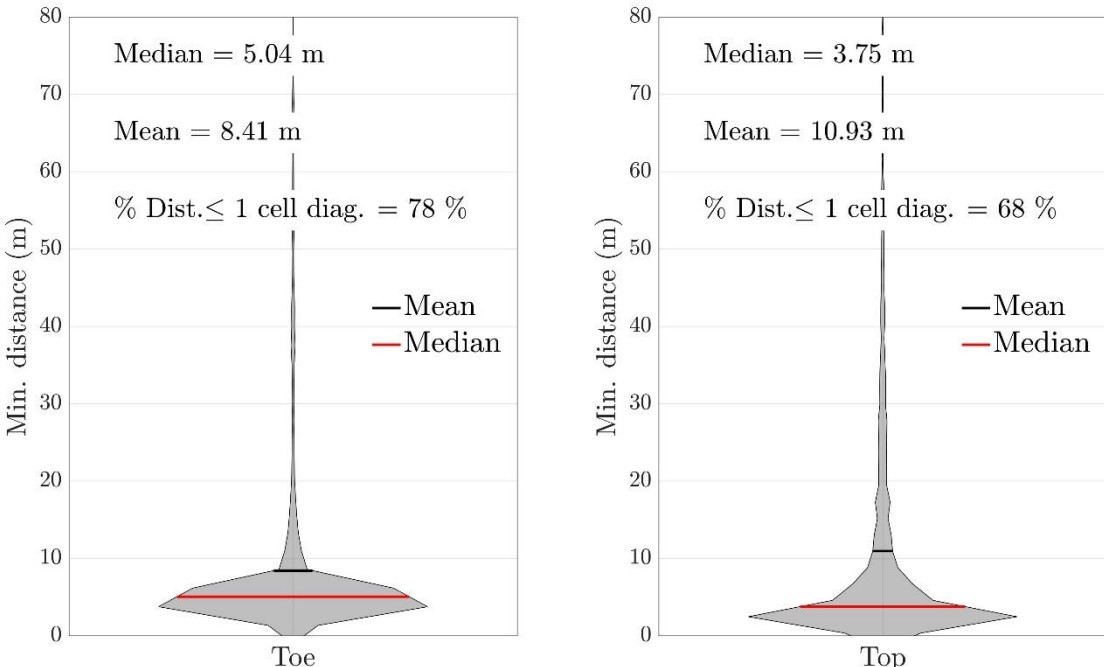

**Figure 11: The proposed method and the PL2016 method outputs are in good agreement. Panels shows the distribution of the minimum distances between cliff toe and top outputs location (for a 5m grid cell, the cell diagonal is 7.07m or the length of the hypotenuse of the square triangle made by two connected sides of the grid cell).**


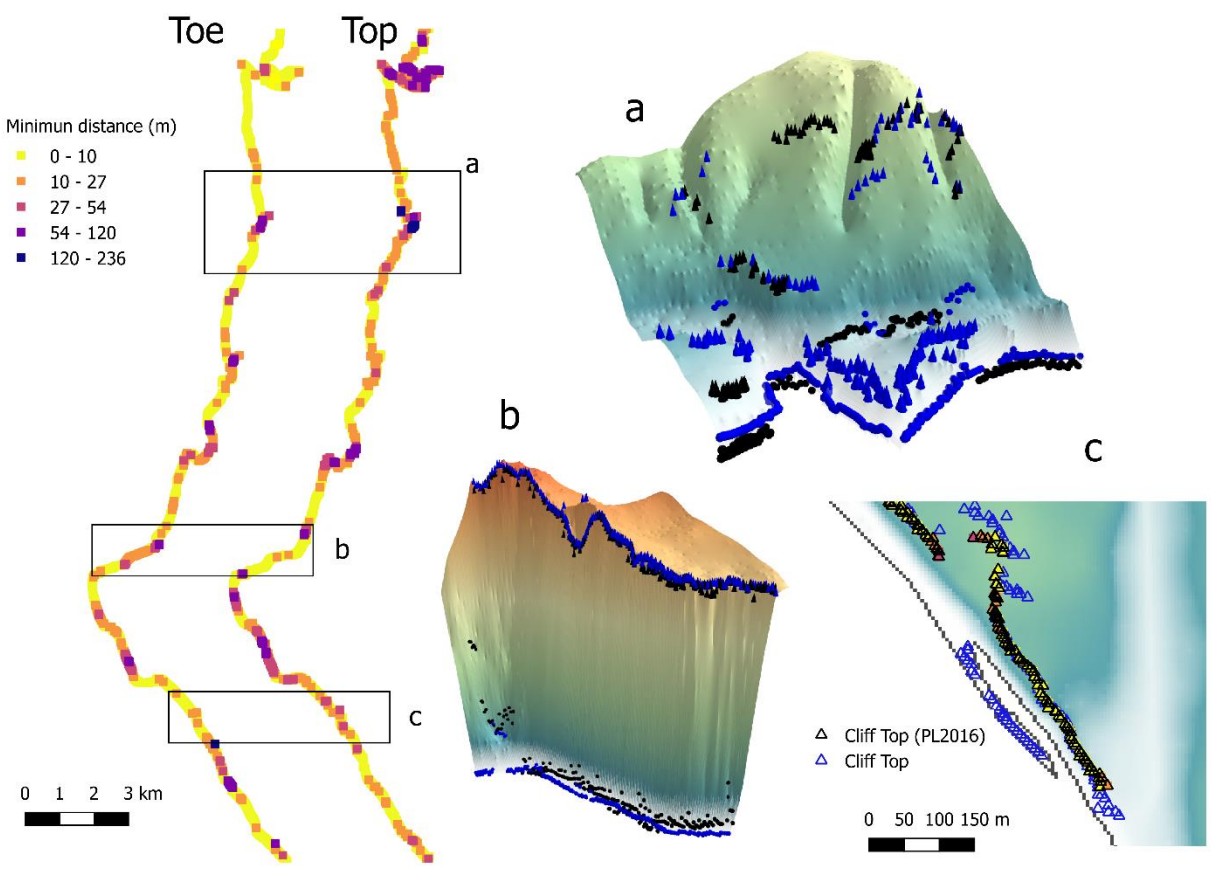

**Figure 12: Minimum distance between the cliff Toe and Top outputs using the PL2016 vs the proposed method. The Toe and Top coloured squares represent the location of the outputs produced by PL2016 and the coloured scale the minimum distance to the outputs produced by this method. Cliff toe/top produced by the proposed method are represented in blue, as spheres/cones for the 3D plots. Panels a, b and shows both model outputs at locations where distances were the greatest. The largest differences between method correspond with (a) where there is a sharp bend on the coast morphology and cliff is not the dominant feature, (b) at the toe of a very steep cliff with a small talus and (c) there is welded bar in front of the cliff .**


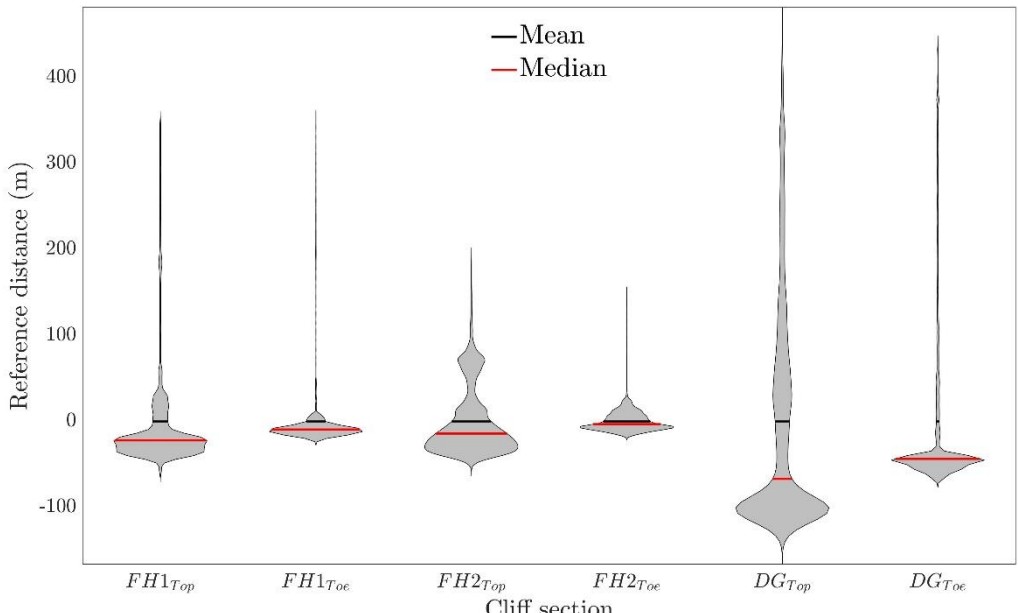

**Figure 13: Distance from participant cliff lines to the mean reference lines for each section.**


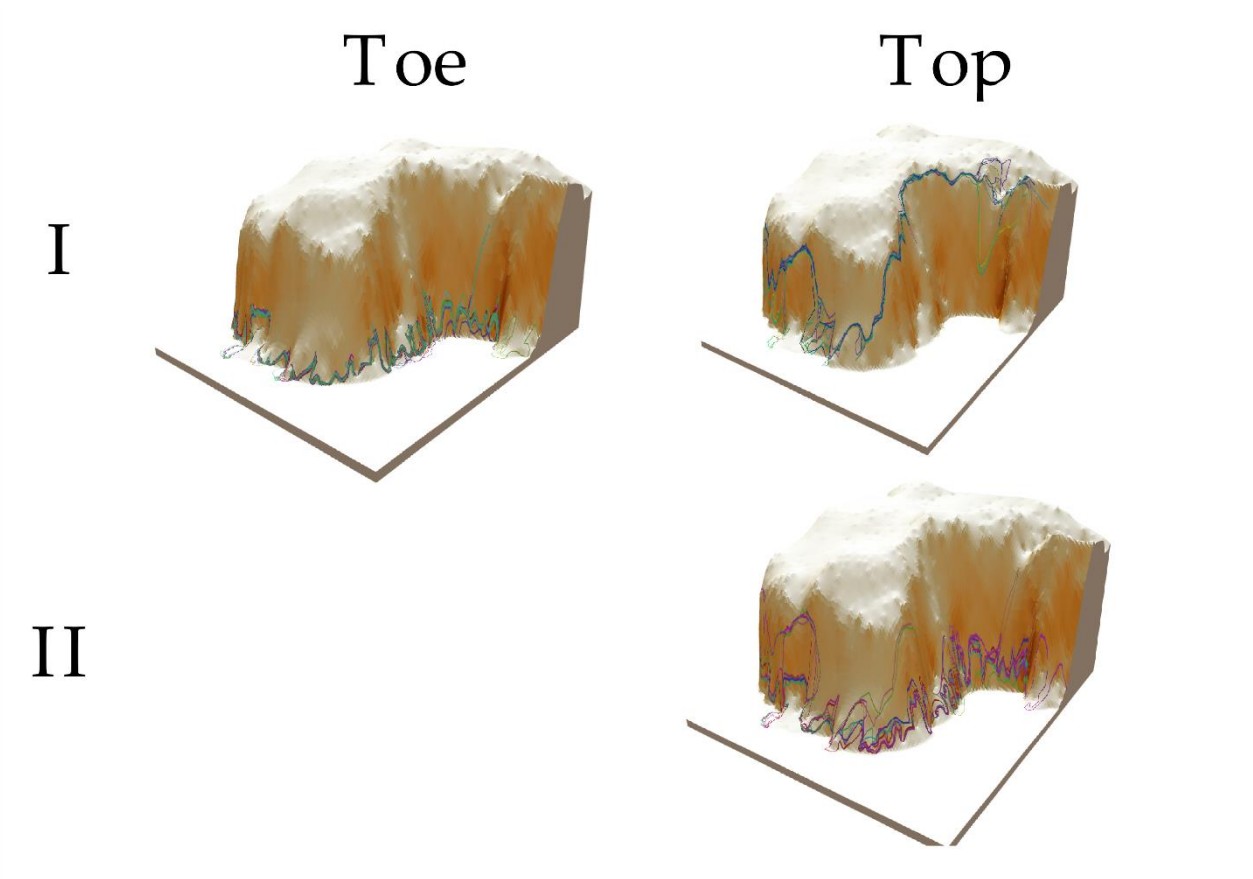

**Figure 14. Hand-digitized cliff top and toe lines (coloured lines) over the DEM for the FH2 site. DEM colours represent slope (darker colours represent higher slopes). The vertical dimension has been exaggerated 10 times. The roman numbers represent the main clusters of hand-digitized lines.**


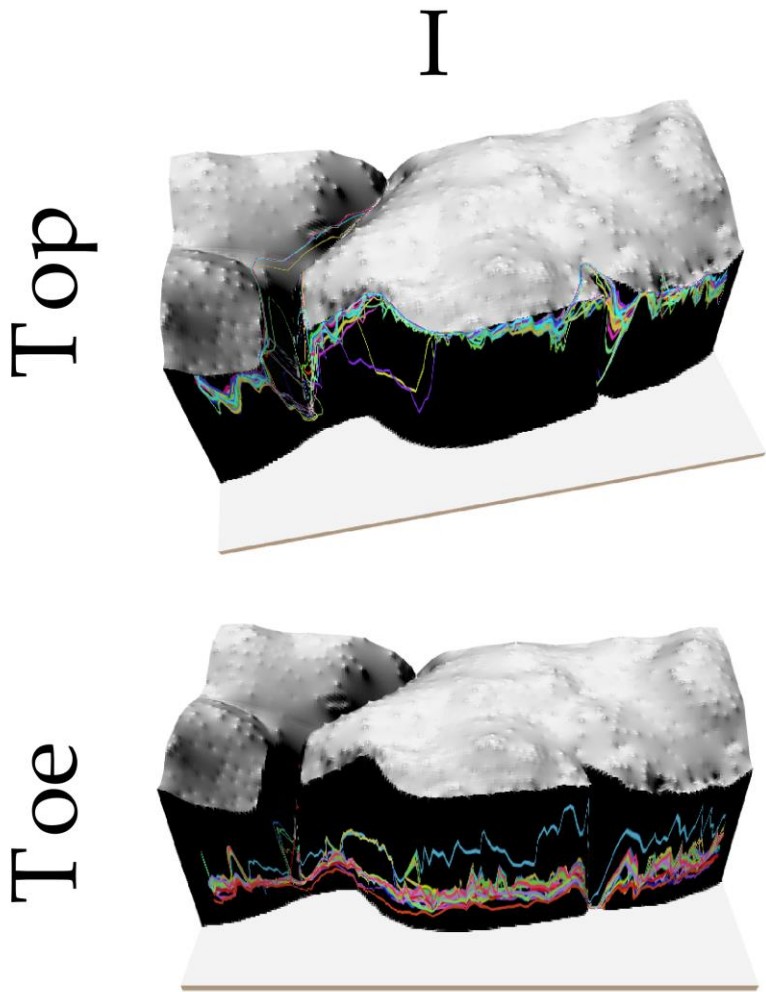

 **Figure 15. Hand-digitized cliff top and toe lines (coloured lines) over the DEM for the FH1 site. DEM colours represent slope (darker colours represent higher slopes). The vertical dimension has been exaggerated 10 times. The roman number represent the main clusters of hand-digitized lines.**

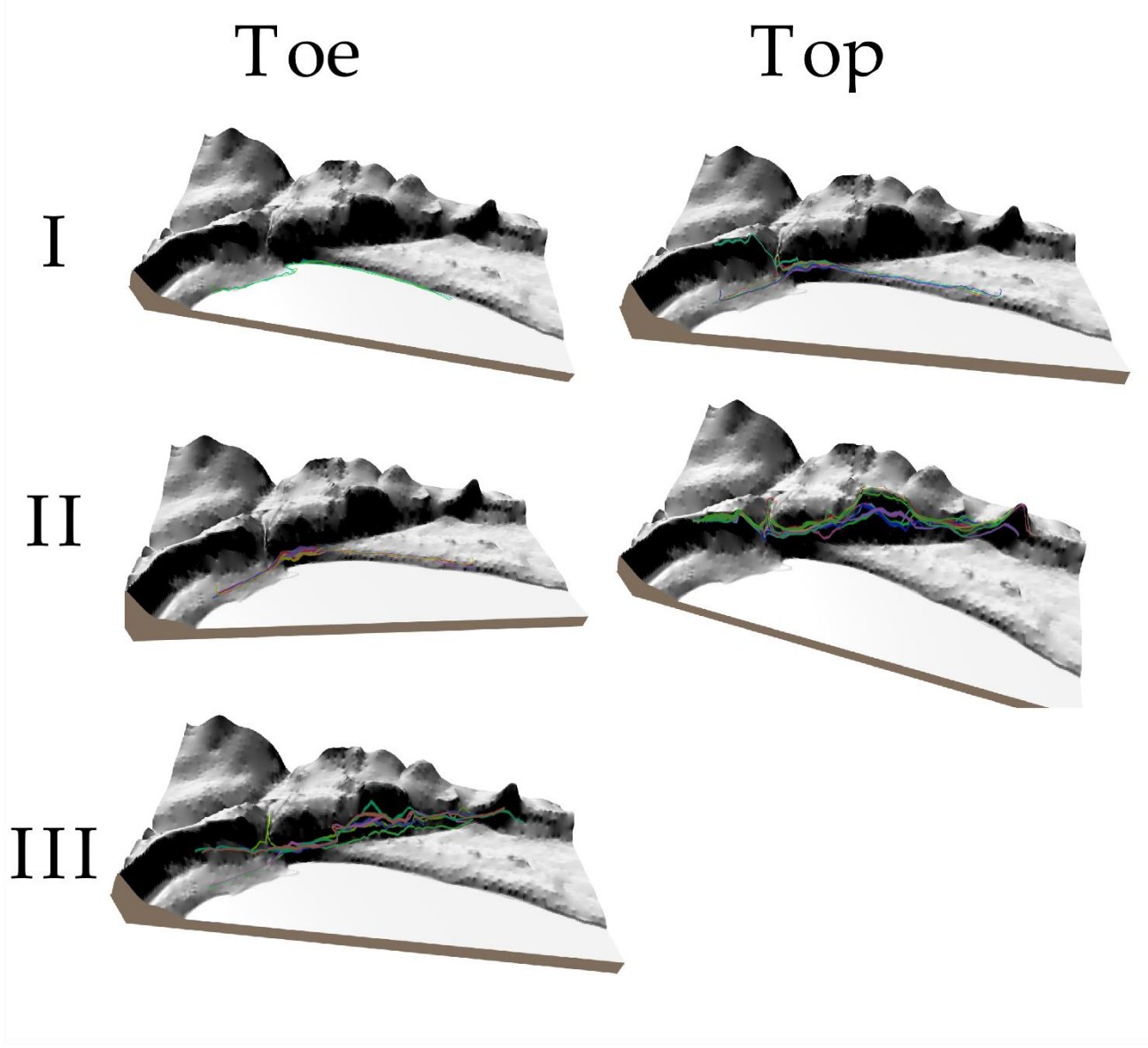


**Figure 16. Hand-digitized cliff top and toe lines (coloured lines) over the DEM for the DG site. DEM colours represent slope (darker colours represent higher slopes). The vertical dimension has been exaggerated 10 times. The roman numbers represent the main clusters of hand-digitized lines.**


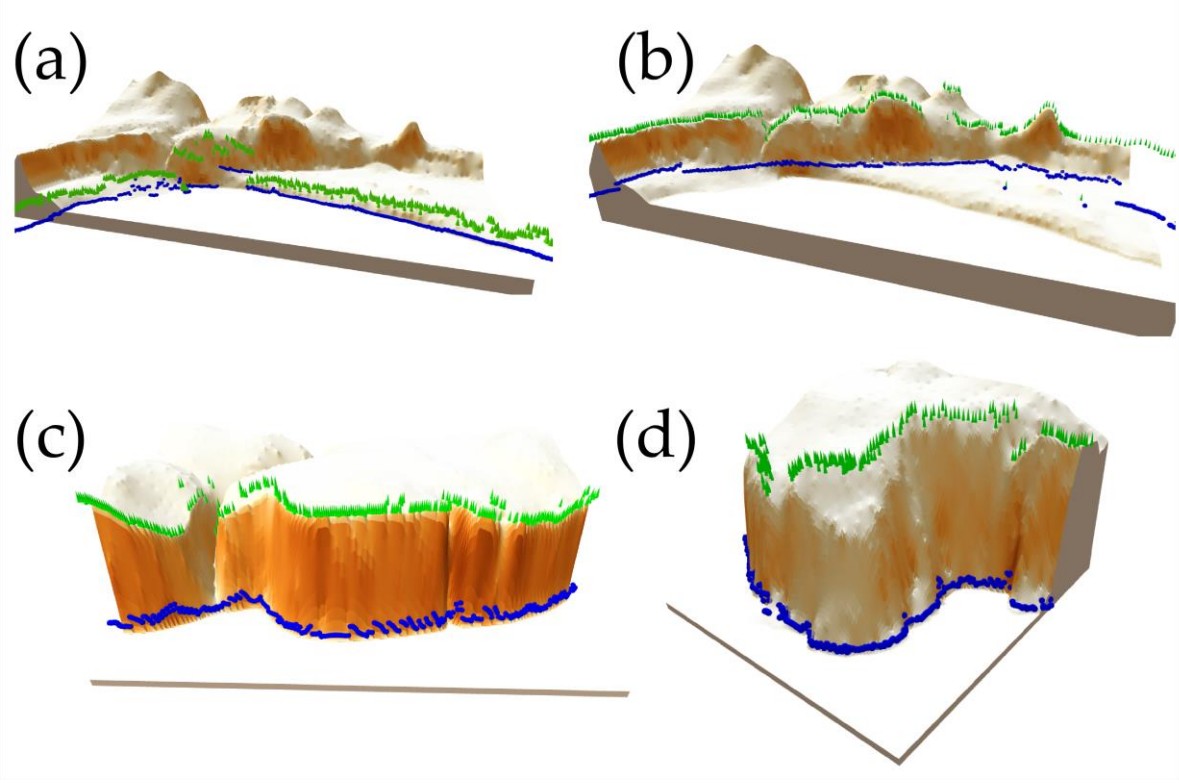

**Figure 17. Automatically delineated cliff top and toe locations (green cone and blue spheres) over the DEM for the DG and FH sites; (a) results using current still water level to delineate the coastline for the DG site; (b) results using a still water level 6m above current level to delineate the Holocene coastline for the DG site; (c) & (d) results using default model setup for FH1 and FH2 respectively. DEM colours represent slope (darker colours represent higher slopes). The vertical dimension has been exaggerated 10 times.**


# Development and technical papers

## Tables captions

**Table 1: Summary of output files produced by the proposed method: name, description and type.**

| Output name | Description | Type |
| --- | --- | --- |
| XX.out | Log of user setup and run performance | ASCII |
| XX.log | Log of simulation run details | ASCII |
| sediment_top_elevation.tif | DEM read by the script and used to delineate the cliff metrics. | GeoTIFF |
| rcoast.tif | Raster coastline. Raster cells that are marked as on the coastline has 1 value and 0 otherwise | GeoTIFF |
| coast_point_XX.shp | Point vector with all the raster coastal points and and four attributes; nCoast is the coast number, nProf is the profile number which is unique for each coastline segment, CoastEl is the elevation in meters of the coast point (i.e. not all coast points have the same elevation but this varies according with the DEM), Chainage or distance in meters in the horizontal plane from the sea point (i.e. it should be 0 m for all coast points by definition). | point Shapefile |
| coast_XX.shp | Point vector with the smoothed coastline. The number of points of coast_XX.shp is equal to the number of points on coast_point_XX.shp | point Shapefile |
| rcoast_normal.tif | Raster coastline normal. Raster cells that are marked as on the coastline-normal has 1 value and 0 otherwise. | GeoTIFF |
| normals_XX.shp | Line vector with the valid coastline normals | line Shapefile |
| invalid_normals_XX.shp | Line vector with the non-valid coastline normals | ine Shapefile |
| coast_nCoast_profile_nProf_XX.csv | CSV file with the elevation profile for profile number "nProf" on coast number "nCoast" and DEM named XX. Each file contains the chainage (i.e. horizontal distance from seaward limit), absolute (x, y) location, elevation above vertical datum and de trended elevation. | ASCII |
| cliff_toe_XX.shp | Point vector with cliff toe position and four attributes; nCoast is the coast number, nProf is the profile number which is unique for each coastline segment, bisOK is a Boolean flag that will be 1 if the profile is valid or 0 otherwise, CliffToeEl is the elevation in meters of the cliff toe, and Chainage of the toe point. | point Shapefile |
| cliff_top_XX.shp | Point vector with cliff top position and four attributes; nCoast is the coast number, nProf is the profile number which is unique for each coastline segment, bisOK is a Boolean flag that will be 1 if the profile is valid or 0 | point Shapefile |

otherwise, CliffTopEl is the elevation of the cliff top, and Chainage of the top point.

XX is the user defined main output and log files name. All elevations are in meters.


**Table 2: Summary of the local sensitivity analysis of cliff toe and top locations to different model set up.**

| | | | **DEM resolution** | | |
|---|---|---|---|---|---|
| | | | **5m** | **10m** | **50m** |
| **Window size smoothing** | | **61pt** | Reference | to DEM resolution only | |
| | | **31pt** | to smoothing only | | |
| | | **7pt** | | | |
| **Vertical threshold** | | **0.5m** | Reference | to DEM resolution, smoothing and threshold all combined | |
| | | **0.01m** | | | |
| | | **1.5m** | to threshold only | | |

Average, standard deviation, maximum, minimum shortest distance between reference and this output

**Table 3: Differences and commonalities of proposed method versus PL2016 method.**

| | Proposed method | PL2016 |
|---|---|---|
| **Differences** | - it is compiled so it is quicker (C++) | - the code is readable so profile extraction function from the DEM along transects is slower (R) |
| | - less pre-processing | - pre-processing work to set up the buffers for generating transects is necessary |
| | - computes only cliff top and toe | - computes secondary inflections on the face of the cliff and if desired identifies the top and 2 toes of a sand bar in front of the cliff (one toe on each side of the sand bar top) |
| | - process concave short profiles (i.e. incomplete cliff profiles look like a check mark) | - reject completely concave profiles (profiles that look like a check mark) |
| | - can deal with very long and narrow promontory by adjusting the normal length automatically | - cannot deal with long and narrow promontory, unless more involved pre-processing is done. |
| | - transects start at a user defined level and projected inland perpendicularly to an automatically delineated smoothed coastline | - transects are projected seaward and inland perpendicularly to a externally delineated coastline |
| **Commonalities** | - after the profile is extracted the 2 codes to extract top and toe are similar using the same logic<br>- both methods output the profile elevation for further processing<br>- rejects short profiles with $N_{min}$ or less elevation points on land, where $N_{min}$ = 3 and 5 for proposed method and PL2016 (there is nothing preventing the methods to be set up for the same $N_{min}$) | |

**Table 4: Cliff toe average, standard deviation, maximum difference and number of samples for the sensitivity analysis to DEM resolution, window size for coastline smoothing and vertical threshold.**

### Cliff TOE

#### Average differences

| Window size | DTM Resolution | | | Vertical threshold | DTM Resolution | | |
|---|---|---|---|---|---|---|---|
| | 5m | 10m | 50m | | 5m | 10m | 50m |
| 61pt | 0 | 4 | 25 | 0.5m | 0 | 4 | 25 |
| 31pt | 1 | 3 | 26 | 0.01m | 1 | 4 | 25 |
| 7pt | 2 | 4 | 25 | 1.5m | 1 | 5 | 26 |

#### Standard deviation

| Window size | DTM Resolution | | | Vertical threshold | DTM Resolution | | |
|---|---|---|---|---|---|---|---|
| | 5m | 10m | 50m | | 5m | 10m | 50m |
| 61pt | 0 | 9 | 38 | 0.5m | 0 | 9 | 41 |
| 31pt | 6 | 4 | 41 | 0.01m | 4 | 7 | 41 |
| 7pt | 13 | 9 | 38 | 1.5m | 9 | 12 | 43 |

#### Maximum differences

| Window size | DTM Resolution | | | Vertical threshold | DTM Resolution | | |
|---|---|---|---|---|---|---|---|
| | 5m | 10m | 50m | | 5m | 10m | 50m |
| 61pt | 0 | 299 | 351 | 0.5m | 0 | 299 | 351 |
| 31pt | 229 | 68 | 282 | 0.01m | 159 | 221 | 351 |
| 7pt | 217 | 204 | 282 | 1.5m | 289 | 299 | 368 |

#### Number of samples

| Window size | DTM Resolution | | | Vertical threshold | DTM Resolution | | |
|---|---|---|---|---|---|---|---|
| | 5m | 10m | 50m | | 5m | 10m | 50m |
| 61pt | | 3213 | 591 | 0.5m | | 3213 | 591 |
| 31pt | 6605 | 3274 | 596 | 0.01m | 6598 | 3213 | 591 |
| 7pt | 6587 | 3279 | 611 | 1.5m | 6598 | 3213 | 591 |

# Development and technical papers

**Table 5: Cliff top average, standard deviation, maximum difference and number of samples for the sensitivity analysis to DEM resolution, window size for coastline smoothing and vertical threshold.**

## Cliff TOP

### Average differences

| Window size | DTM Resolution | | | Vertical threshold | DTM Resolution | | |
| --- | --- | --- | --- | --- | --- | --- | --- |
| | 5m | 10m | 50m | | 5m | 10m | 50m |
| 61pt | 0 | 6 | 32 | 0.5m | 0 | 6 | 33 |
| 31pt | 3 | 5 | 32 | 0.01m | 0 | 6 | 31 |
| 7pt | 5 | 8 | 29 | 1.5m | 8 | 6 | 37 |

### Standard deviation

| Window size | DTM Resolution | | | Vertical threshold | DTM Resolution | | |
| --- | --- | --- | --- | --- | --- | --- | --- |
| | 5m | 10m | 50m | | 5m | 10m | 50m |
| 61pt | 0 | 11 | 62 | 0.5m | 0 | 11 | 33 |
| 31pt | 10 | 21 | 63 | 0.01m | 2 | 11 | 31 |
| 7pt | 19 | 33 | 59 | 1.5m | 8 | 14 | 37 |

### Maximum differences

| | DTM Resolution | | | Vertical threshold | DTM Resolution | | |
| --- | --- | --- | --- | --- | --- | --- | --- |
| | 5m | 10m | 50m | | 5m | 10m | 50m |
| 61pt | 0 | 186 | 470 | 0.5m | 0 | 186 | 470 |
| 31pt | 153 | 477 | 479 | 0.01m | 54 | 186 | 458 |
| 7pt | 465 | 497 | 502 | 1.5m | 202 | 186 | 470 |

### Number of samples

| Window size | DTM Resolution | | | Vertical threshold | DTM Resolution | | |
| --- | --- | --- | --- | --- | --- | --- | --- |
| | 5m | 10m | 50m | | 5m | 10m | 50m |
| 61pt | | 3213 | 591 | 0.5m | | 3213 | 591 |
| 31pt | 6605 | 3274 | 596 | 0.01m | 6598 | 3213 | 591 |
| 7pt | 6587 | 3279 | 611 | 1.5m | 6598 | 3213 | 591 |

**Table 6: ASCII input file with the user defined delineation parameters.**


```
; SIMPLE TEST DATA
;
; Run information --------------------------------------------------------------------------------------
1 Main output/log file names              [omit path and extension]: dg
2 DTM file  (DTM MUST BE PRESENT)                  [path and name]: in/DG/DG.tif;
3 Still water level (m)  used to find the shoreline            : 1.0     ;
4 Coastline smoothing        [0=none, 1=running mean, 2=Savitsky-Golay]: 1
5 Coastline smoothing window size              [must be odd]: 61              ; was 205 for S-G
6 Polynomial order for Savitsky-Golay coastline smoothing          [2 or 4]: 4

; If user wants to use a given shoreline vector instead of extracting it from the DTM
7 Shoreline shape file (OPTIONAL GIS FILES)            [path and name]:

; Advance Run information ------------------------------------------------------------------------------
8 GIS raster output format              [blank=same as DEM input]: gtiff          ; gdal-config --formats for others
9   If needed, also output GIS raster world file?            [y/n]: y
10   If needed, scale GIS raster output values?            [y/n]: y
11 GIS vector output format                  : ESRI Shapefile         ; ogrinfo --formats for others

12 Random edge for coastline search?              [y/n]: y
13 Random number seed(s)                  : 280761
14 Length of coastline normals (m)              : 500     ; was 80
15 Vertical tolerance to avoid false cliff top/toes (m)              : 0.5
;  END  OF  FILE  -----------------------------------------------------------------------------------------10      If  needed,  scale  GIS  raster  output  values?
[y/n]: y
```