# Peer review of "Development of an automatic delineation of cliff top and toe on very irregular planform coastlines (CliffMetrics v1.0)"

_Geoscientific Model Development, 2018_

## Short Comment (SC1) · 22 Jun 2018

To maintain reproducibility the authors need to tag the release in the GitHub repository that matches the model version described in the paper. This grantees that check-ins after manuscript submission don't compromise reproducibility. As explained in https://www.geoscientific-model-development.net/about/manuscript_types.html the preferred reference to this release is through the use of a DOI which is then cited in the paper. For projects in GitHub a DOI for a released code version can easily be created using for instance Zenodo, see https://guides.github.com/activities/citable-code/ for details.

Lutz Gross GMD Executive Editor

---

## Referee Comment (RC1) · Anonymous Referee #1 · 4 Jul 2018

This is a very interesting paper and the way it is presented is very useful. I particularly like the inclusion of the code at the end. The paper is well written - a few minor edits to sentences would tidy it up a little - and well structured, and the illustrations are useful. For me the text is a little wordy in places - and whilst I applaud the way the discussion is integrated with aspects of the method - I would like to have seen the method completely separated from any discussion etc. and perhaps the inclusion of a flow diagram of the method added. But this may just be a personal view. The only other thing that I wondered about is whether there is any thoughts on using a UAV/drone based model as another comparison if appropriate. Overall this should be of benefit to a number of people.

---

## Referee Comment (RC2) · Anonymous Referee #2 · 22 Aug 2018

This paper presents a cliff top and toe delineation method that is much quicker than existing manual or semi-automated approaches. The work and code are a useful contribution, however the setup of the sensitivity analysis and comparison to the hand-drawn maps lacks detail. Analysis methods are not always justified sufficiently for the reader. Below are some specific comments related to these points.

Line 125, presumably LiDAR could also be used, but more widely available data sets such as global DEM (ASTER, SRTM) might have insufficient quality? It would be interesting to comment on DEM source if they authors have any insights because NEXTMap is UK focused?

Line 140: Is this something that could be illustrated in the figure? As presented it's obviously not possible to see the effect of changing resolution from 5 - 50m. Also, if the resolution is changed from 5 to 50 m how can the cliff top shift by only 10 m where the resolution is only 50. Does this not imply that at least at some locations the cliff top and bottom will have to move by 25 m?

Section 2.1 Why not use the same locations for the sensitivity analysis and hand drawn analysis? Please justify the choice of sites and decision to use different sites for the various analysis

Line 164: Would it be possible to define seed locations on islands to avoid having to manipulate the input DEM? Also do you have islands in your test cases, the text was slightly vague is this regard?

Line 200: please explain and justify your choice of sensitivity analysis method. It seems that you have done a one-at-a-time analysis rather than used any of the more advanced method available in the referenced toolbox (I'm not sure from the text)? Presumably interactions between the parameters are not thought to be important?

Line 211: Is NEXTMap accurate to 0.5 m in steep areas? I'd have thought less accurate, could you give a citation for this or consider increasing the range of the parameter if its linked to DEM quality. Could your maximum threshold be below the accuracy of the DEM in some steep locations?

Section 3.3: Here I was expecting some form of test that the automatically derived line was not statistically different to the hand drawn ensemble. This looks like it might be the case of the FH 1&2 toe but is less clear in the other cases, especially at DG site where the model-based estimates appear to have diverged from the hand drawn ensemble (if I have interpreted the plots correctly). Its half done in the discussion but only the variances are compared not the ensembles. Furthermore, would different parameterisations of the model have produced top or toe delineations closer to the hand drawn ensemble mean and does the model also flip between different cliff lines

when given different parameters, if not why not?. There is some discussion of the hand drawn results in the discussion section but for me insufficient discussion on the model behaviour. How did you decide that the sensitivity of model outputs was less than the hand digitised uncertainty where there is no specific analysis of this in section 3.3? Is less sensitivity desirable if you do not capture the same ensemble mean (assuming hand delineation is the accepted benchmark)? There is no discussion of this point.

Section 4: discussion: A preamble is needed for the discussion to set up the reader.

Finally, in the discussion are the variances reported for the sensitivity analysis from a different site (St Bees) to the hand drawn maps (Sandhead & Flamborough head). If so this comparison is completely invalid and the sensitivity analysis is needed at the hand drawn sites? Even if this is not the case it does read as if two separate analysis have been conducted and put together rather than a progression of analysis towards a complete picture of the methods suitability.

---

## Author Comment (AC1) · 10 Sep 2018

We have now created a DOI of the release associated with the manuscript using Zenodo.

The DOI URL is https://doi.org/10.5281/zenodo.1412486

On the description of the release version we have now made it clear that this is the software version associated with this manuscript (see description below and by linking the URL above)

Description of First release associated with the manuscript:

[Figure]

This is the first release of the CliffMetrics algorithm for automatic delineation of cliff top and toe. This is the software version used for the Geoscientific Model Development manuscript "Development of an automatic delineation of cliff top and toe on very irregular planform coastlines (CliffMetrics v1.0)"

https://doi.org/10.5194/gmd-2018-83

gmd-2018-83.pdf

---

## Author Comment (AC2) · 17 Sep 2018

The authors of this manuscript would like to thank the anonymous referee #1 for his/her review. We have numbered the main comments from the reviewer as shown below: 1. The paper is well written - a few minor edits to sentences would tidy it up a little - and well structured, and the illustrations are useful. 2. For me the text is a little wordy in places - and whilst I applaud the way the discussion is integrated with aspects of the method - I would like to have seen the method completely separated from any discussion etc. and perhaps the inclusion of a flow diagram of the method added. But this may just be a personal view. 3. The only other thing that I wondered about is whether

there is any thoughts on using a UAV/drone based model as another comparison if appropriate.

The authors' response to each one of the reviewers comment are summarized below. Two versions of the revised manuscript has been submitted together with the response to the reviewers' comments: one version with all changes highlighted using Track-changes and a final version (i.e. all changes accepted) revised manuscript. 1. The whole manuscript has been proof read and sentences has been streamlined when possible. 2. A flow diagram of the method has been added and any discussion on the method section has been re-allocated to the discussion section. 3. The proposed algorithm is sensitive to the resolution of the DEM but agnostic regarding the origin of the DEM. UAV/drone based DEM are therefore also valid sources of DEM. We have stated this explicitly in the discussion section of the revised manuscript.

Please also note the supplement to this comment:
https://www.geosci-model-dev-discuss.net/gmd-2018-83/gmd-2018-83-AC2-supplement.zip

---

## Author Comment (AC3) · 17 Sep 2018

The authors of this manuscript would like to thank the anonymous referee #2 for his/her review.

We have numbered the main comments from the reviewer as shown below: 1. This paper presents a cliff top and toe delineation method that is much quicker than existing manual or semi-automated approaches. The work and code are a useful contribution, however the setup of the sensitivity analysis and comparison to the hand-drawn maps lacks detail. Analysis methods are not always justified sufficiently for the reader. 2. Line 125, presumably LiDAR could also be used, but more widely available data sets such

as global DEM (ASTER, SRTM) might have insufficient quality? It would be interesting to comment on DEM source if they authors have any insights because NEXTMap is UK focused? 3. Line 140: Is this something that could be illustrated in the figure? As presented it's obviously not possible to see the effect of changing resolution from 5 - 50m. Also, if the resolution is changed from 5 to 50 m how can the cliff top shift by only 10 m where the resolution is only 50. Does this not imply that at least at some locations the cliff top and bottom will have to move by 25 m? 4. Section 2.1 Why not use the same locations for the sensitivity analysis and hand drawn analysis? Please justify the choice of sites and decision to use different sites for the various analysis 5. Line 164: Would it be possible to define seed locations on islands to avoid having to manipulate the input DEM? Also do you have islands in your test cases, the text was slightly vague is this regard? 6. Line 200: please explain and justify your choice of sensitivity analysis method. It seems that you have done a one-at-a-time analysis rather than used any of the more advanced method available in the referenced toolbox (I'm not sure from the text)? Presumably interactions between the parameters are not thought to be important? 7. Line 211: Is NEXTMap accurate to 0.5 m in steep areas? I'd have thought less accurate, could you give a citation for this or consider increasing the range of the parameter if its linked to DEM quality. Could your maximum threshold be below the accuracy of the DEM in some steep locations? 8. Section 3.3: Here I was expecting some form of test that the automatically derived line was not statistically different to the hand drawn ensemble. This looks like it might be the case of the FH 1&2 toe but is less clear in the other cases, especially at DG site where the model-based estimates appear to have diverged from the hand drawn ensemble (if I have interpreted the plots correctly). Its half done in the discussion but only the variances are compared not the ensembles. Furthermore, would different parameterisations of the model have produced top or toe delineations closer to the hand drawn ensemble mean and does the model also flip between different cliff lines when given different parameters, if not why not?. There is some discussion of the hand drawn results in the discussion section but for me insufficient discussion on the model behaviour. How

did you decide that the sensitivity of model outputs was less than the hand digitised uncertainty where there is no specific analysis of this in section 3.3? Is less sensitivity desirable if you do not capture the same ensemble mean (assuming hand delineation is the accepted benchmark)? There is no discussion of this point. 9. Section 4: discussion: A preamble is needed for the discussion to set up the reader. 10. Finally, in the discussion are the variances reported for the sensitivity analysis from a different site (St Bees) to the hand drawn maps (Sandhead & Flamborough head). If so this comparison is completely invalid and the sensitivity analysis is needed at the hand drawn sites? Even if this is not the case it does read as if two separate analysis have been conducted and put together rather than a progression of analysis towards a complete picture of the methods suitability.

The authors' response to each one of the reviewers comment are summarized below. Two versions of the revised manuscript has been submitted together with the response to the reviewers' comments: one version with all changes highlighted using Track-changes and a final version (i.e. all changes accepted) revised manuscript. 1. In the revised manuscript, the setup of the sensitivity analysis and comparison to the hand-drawn maps has been explained in more detail as outlined on the answer to the specific comments shown below. 2. The only requirement of the proposed method regarding the DEM is that it should include the cliff toe and top (i.e. cover from the shoreline to sufficiently far inland to capture the cliff top). The algorithm is agnostic regarding the method used to collect the data (i.e. air-prone-Radar, terrestrial/UAV LiDAR, ...). We have used DEM from UK as an example of very irregular plan-shape coastlines but the method is in principle transferable to any other DEM. This requirement has now been explicitly stated on the methodology section. We have shown how the cliff top and toe delineation might varies with DEM resolution but the question regarding if the quality of existing global DEM is sufficient is likely to be specific to the delineation purpose and out of the scope of this work. 3. The difference on the cliff toe/top locations due to changes on the DEM resolution is illustrated on Figure 7 and Table 4. As shown in Table 4, average differences due

to a resolution change from a 5m DEM to a 50m DEM are, as rightly pointed by the reviewer, on average of 25m. The sentence at line 140 starting as "A visual inspection . . ."is therefore somehow misleading and has been removed from the revised version. 4. This is a good and key point. The following text has been added to the study site selection to clarify the rationale of the site selection: "The aims of the sensitivity, model-to-model comparison and hand-digitized analysis are different and therefore the places selected to conduct each analysis are different too. Our sensitivity analysis and model-to-model comparison investigates the way in which the variation in the output can be attributed to variations in the different input factors (Pianosi et al., 2016) or different automatic delineation procedures respectively. The hand-digitized analysis illustrates the importance of the data outputs screening and algorithm behaviour. For the sensitivity analysis and model-to-model comparison, we have focused on a coastal cliff-dominated region with irregular plan-shape to make our findings more transferable to other similar cliffed coastlines elsewhere. For the hand-digitized analysis we have selected a challenging coastal region (i.e. very irregular shape, complex cliff profile sections intercalated with non-cliffed sections) to highlight the importance of screening the results and running the algorithm iteratively until the hand-digitized and automatically delineated cliff top and toe locations converges." 5. Cropping the DEM to PC-manageable-sizes is an unavoidable requirement. The requirement of been careful ensuring that the islands-inland-topography do cross the cropped tiles is very straight forward. In this context, we have not explored more elaborated methods of identifying automatically island within the tile. A flow chart has been included on the revised manuscript to make clear the minimum pre-processing required. 6. See response to question #4 7. NextMap Britain Horizontal accuracies are +/-2.5m horizontal on slopes less than 20 degrees. Accuracies on steeper slopes is not provided on the NextMap specifications. We have now included the important point of horizontal accuracies provided been for slope less than 20 degrees. The maximum threshold elevation is relative to the detrended elevation and it can be smaller than the DEM resolution. This is also now clearly stated in the revised manuscript. 8.

Section 3.3 has been re-edited on the revised manuscript. Old Figure 13 showing the hand-digitized and automatic delineated profiles has been replaced by several figures. We now more clearly show that hand-digitized cliff edges seems to be biased towards slope changes rather than maximum and minimum elevation profiles making the one to one comparison not possible. Following the reviewer advice we have nevertheless shown how the algorithm is able to differentiate between the active and Holocene profile for the DG site and how the same set up used, as a reference for the sensitivity analysis seems to produce realistic cliff top and toe locations. 9. A preamble is now included on the discussion section as suggested. 10. See response to question # 8. The text below has also been added to the discussion section on the revised manuscript: "Hand-digitized cliff top and toe location spread between participants are of the order of 4 to 23 diagonal cells (i.e. for a DEM of 5m cell size). These large differences seems to be driven by the bias towards using changes of slope as the preferred cliff top and toe locations when hand digitizing over an aerial photography. This bias prevented a model to hand-digitized more in detail comparison. We have shown how the inputs parameter can be modified to resolve both active and Holocene cliff lines. The algorithm reference set up seems to be robust enough for the two FH sites despite the difference on the plan shape at both sites. Our algorithm delineates the cliff top and toe and produce all model outputs for a 1km section of coast in less than one second while hand-digitizing the same length of coast took around 10 minutes. Thus our algorithm is about 5 orders of magnitude faster than hand digitizing."

Please also note the supplement to this comment:
https://www.geosci-model-dev-discuss.net/gmd-2018-83/gmd-2018-83-AC3-supplement.zip